# Experience-dependent weakening of callosal synaptic connections in the absence of postsynaptic FMRP

Zhe Zhang, Jay R Gibson*, Kimberly M Huber*

Department of Neuroscience, O'Donnell Brain Institute, University of Texas Southwestern Medical Center, Dallas, United States

**Abstract** Reduced structural and functional interhemispheric connectivity correlates with the severity of Autism Spectrum Disorder (ASD) behaviors in humans. Little is known of how ASD-risk genes regulate callosal connectivity. Here, we show that *Fmr1*, whose loss-of-function leads to Fragile X Syndrome (FXS), cell autonomously promotes maturation of callosal excitatory synapses between somatosensory barrel cortices in mice. Postnatal, cell-autonomous deletion of *Fmr1* in postsynaptic Layer (L) 2/3 or L5 neurons results in a selective weakening of AMPA receptor- (R), but not NMDA receptor-, mediated callosal synaptic function, indicative of immature synapses. Sensory deprivation by contralateral whisker trimming normalizes callosal input strength, suggesting that experience-driven activity of postsynaptic *Fmr1* KO L2/3 neurons weakens callosal synapses. In contrast to callosal inputs, synapses originating from local L4 and L2/3 circuits are normal, revealing an input-specific role for postsynaptic *Fmr1* in regulation of synaptic connectivity within local and callosal neocortical circuits. These results suggest direct cell autonomous and postnatal roles for FMRP in development of specific cortical circuits and suggest a synaptic basis for long-range functional underconnectivity observed in FXS patients.

**\*For correspondence:**
Jay.Gibson@UTSouthwestern.
edu (JRG);
Kimberly.Huber@
utsouthwestern.edu (KMH)

**Competing interest:** The authors declare that no competing interests exist.

## Introduction

Disrupted structural and functional brain connectivity has been widely observed in patients with autism spectrum disorder (ASD) (*Dimond et al., 2019*; *Holiga et al., 2019*; *Rane et al., 2015*). A common finding in ASD is reduced corpus callosum integrity and interhemispheric functional connectivity, the latter of which correlates with autistic symptoms (*Li et al., 2019*; *O'Reilly et al., 2017*; *Yao et al., 2021*). The corpus callosum connects bilateral hemispheres and functions to synchronize cortical circuits necessary for sensory-motor processing, attention, perception and higher cognitive functions (*Schulte and Müller-Oehring, 2010*). Little is known of how autism-risk genes regulate development of callosal connectivity and the cellular or synaptic basis of reduced functional connectivity in ASD.

To provide insight into these questions, we have studied the role of the Fragile X Mental Retardation gene (*Fmr1*) in development of callosal synaptic connections in mice. Loss-of-function mutations in *FMR1* in humans cause Fragile X Syndrome (FXS), the most common inherited form of intellectual disability and leading monogenic cause of ASD (*Garber et al., 2008*; *Niu et al., 2017*). Children, age 6–24 months, with FXS have reduced structural integrity of white matter tracts, including the corpus callosum (*Swanson et al., 2018*). *Fmr1* knockout (KO) mice, an animal model for FXS, have a similar reduction in white matter tract integrity as well as decreased functional coherence among different cortical regions as measured with functional MRI (*Haberl et al., 2015*; *Zerbi et al., 2018*). Specifically, neural networks involved in sensory processing show severe functional underconnectivity, including both intra- and inter-hemispheric cortical circuits (*Zerbi et al., 2018*). The cellular or synaptic basis for decreased inter-region functional coherence in FXS is unknown, and whether this is due to direct or indirect roles for *Fmr1* in cortical neurons is unclear.

In addition to reduced long-range connectivity, there are reports of increased connectivity and hyperexcitability of local cortical circuits in humans with ASD and FXS (*Ciarrusta et al., 2020*; *Courchesne and Pierce, 2005*). In the *Fmr1* KO mouse, there is strong evidence for hyperexcitable local cortical circuits , including in visual, auditory, and somatosensory cortices (*Contractor et al., 2015*; *Gibson et al., 2008*; *Gonçalves et al., 2013*; *Hays et al., 2011*; *Osterweil et al., 2013*). Hyperactive cortical circuits are also observed in humans with FXS, and *Fmr1* KO mice as an increase in resting state gamma power in the resting state EEG (*Jonak et al., 2020*; *Lovelace et al., 2018*; *Wang et al., 2017*). Multiple cellular and synaptic alterations likely contribute to hyperexcitability of local circuits including synaptically hyperconnected pyramidal neurons, reduced inhibitory neuron activity and changes in intrinsic excitability (*Gibson et al., 2008*; *Goel et al., 2018*; *He et al., 2014*; *Zhang et al., 2014*). Layer (L) 2/3 and L5 cortical pyramidal neurons receive and integrate excitatory synaptic inputs from homotopic contralateral hemisphere (callosal) as well as other long-range inputs from ipsilateral cortical regions and local cortical circuits. Little is known if or how the development of local and long-range synaptic inputs is balanced or if this balance is regulated by ASD-risk genes.

*Fmr1* encodes Fragile X Mental Retardation Protein (FMRP), an RNA-binding protein that interacts with many mRNAs including those encoding pre- and post-synaptic proteins (*Darnell et al., 2011*). It is perhaps through this diversity of mRNA targets that FMRP regulates multiple properties of synapses, including maturation, pruning and acute forms of synaptic plasticity (*Huang et al., 2013*; *Pfeiffer and Huber, 2009*). Regarding excitatory cortical synapses in somatosensory cortex, *Fmr1* regulates maturation of thalamocortical inputs to L4 neurons as well as between local cortical circuits. A common finding is a delayed maturation of excitatory synapses on *Fmr1* KO neurons, as observed by the delayed presence of NMDA receptor-only, or 'silent', synapses and acquisition of AMPAR-mediated synaptic transmission at thalamocortical synapses onto L4 neurons and between locally connected L5 neurons (*Contractor et al., 2015*; *Patel et al., 2014*). Furthermore, L4 to L2/3 synaptic inputs are weak and delayed in their developmental strengthening, and dendritic spines on cortical pyramidal neurons are thin and filopodial-like, resembling immature spines (*Cruz-Martín et al., 2010*; *Huang et al., 2013*; *Nimchinsky et al., 2001*). L5 local synaptic connections ultimately mature and appear normal during the 4th postnatal week but then fail to prune, which results in hyperconnectivity of L5 neurons at 4–5 weeks of age as compared to wild-type cortex (*Patel et al., 2014*). Importantly, the delayed development and hyperconnectivity of L5 circuits are due to a cell autonomous, postsynaptic role for FMRP in L5 neurons, suggesting a direct role of FMRP in coordinating multiple synapse development processes. The cellular locus of FMRP function in development of L4 to L2/3 inputs or whether postsynaptic FMRP similarly regulates development of long-range synaptic connections is unknown.

Using viral mediated expression of Channelrhodopsin 2 (ChR2) in callosal projecting cortical neurons (*Petreanu et al., 2009*), we observe weak callosal synaptic inputs onto L2/3 and L5 *Fmr1* KO neurons that are mediated by a cell autonomous, postsynaptic, and postnatal role of FMRP. Callosal inputs have a selective deficit in AMPA receptor (R), but not NMDAR-, mediated synaptic transmission, indicative of synapse maturation deficit in *Fmr1* KO neurons. Sensory deprivation by whisker trimming normalized callosal input strength suggesting that sensory experience-driven activity of postsynaptic *Fmr1* KO neurons weakens callosal synapses. Surprisingly, local excitatory inputs were normal on L2/3 neurons with postsynaptic *Fmr1* deletion, revealing differential regulation of local and callosal synaptic connections by FMRP. These results reveal a cellular and synaptic substrate for reduced interhemispheric connectivity in FXS as well as imbalanced activity with local circuits.

## Results

### Optogenetic activation of callosal axons shows weak excitatory synaptic inputs onto L2/3 pyramidal neurons in barrel cortex of *Fmr1* KO mice

To measure callosal synaptic function and connectivity between hemispheres, mCherry-tagged Channelrhodpsin-2 (ChR2) was expressed in callosal projecting cortical neurons by injecting AAV9 into in one hemisphere of primary somatosensory cortex of postnatal day (P) one pups. At P18-25, acute coronal slices containing barrel cortex contralateral to the AAV injected side were prepared and whole cell patch clamp recordings of L2/3 pyramidal neurons were performed in the region innervated by

fluorescently labeled callosal axons (*Figure 1A–B*). Monosynaptic excitatory postsynaptic currents (EPSCs) were elicited by stimulating ChR2 expressing callosal axons with brief (2 ms) blue LED light pulses in the presence of tetrodotoxin (TTX) and 4-aminopyridine (4-AP) as described previously (*Petreanu et al., 2007*; *Rajkovich et al., 2017*; *Figure 1C*). The blue LED was delivered through a 40 X lens centered around the soma of approximately 350 μm diameter in size and thus the amplitude of LED-evoked EPSCs likely reflects the overall strength of callosal synaptic inputs onto the recorded neuron. The amplitude of LED-evoked EPSCs in *Fmr1* KO neurons was reduced by ~40 % compared to WT. Similarly, reduced EPSC amplitudes were observed by comparing raw values or when normalized to LED power (*Figure 1D*). To determine if weak callosal synaptic inputs were localized to a subcellular region on recorded L2/3 neurons (e.g. apical vs. distal dendrites), we utilized a method termed subcellular Channelrhodopsin2-assisted circuit mapping (sCRACM *Petreanu et al., 2009*). A blue laser was flashed across an array of locations, relative to the soma of the recorded neuron, in a pseudorandom order to elicit glutamate release from ChR2-expressing callosal axon terminals and EPSCs at localized dendritic sites (*Figure 1E–F*). The mean amplitude of EPSCs at each site was converted into a color map for each recorded neuron and these maps were then aligned by the position of soma and averaged within genotype. The resulting average map depicts the subcellular distribution of callosal synaptic input strengths onto WT or *Fmr1* KO neurons (*Figure 1G*). Results reveal that callosal synaptic inputs onto WT L2/3 neurons are strongest at the proximal apical dendrites (*Figure 1H*), consistent with previous reports (*Petreanu et al., 2009*) and there is an interaction of *Fmr1* and vertical position (*p < 0.05, F(15, 849) = 1.723; mixed-effects ANOVA). Comparison of mean EPSC amplitudes from proximal apical dendrites reveals a 50 % reduction in callosal synaptic input strength in *Fmr1* KO mice (*Figure 1H*).

## Postsynaptic *Fmr1* cell-autonomously promotes callosal synapses

FMRP is known to have both pre- and postsynaptic, cell autonomous roles in excitatory synapse development, depending on the cell type and synaptic input (*Patel et al., 2013*; *Patel et al., 2014*; *Pfeiffer et al., 2010*). To determine the cell-autonomous and synaptic locus of FMRP function in development of callosal synapses, we deleted *Fmr1* in a sparse population (~3%–5%) (*Figure 2—figure supplement 1*) of postsynaptic L2/3 pyramidal neurons by injecting AAV expressing GFP-tagged Cre (AAV9. GFP-Cre) into the left lateral ventricle of P1 pups (*Kim et al., 2013*) with a floxed *Fmr1* gene (*Fmr1*^fl/fl or *Fmr1*^fl/y) (*Figure 2A*; *Mientjes et al., 2006*). In the same animals, AAV.ChR2-mCherry was injected into the right barrel cortex to label callosal axons as in *Figure 1*. This experimental design allowed us to measure the cell autonomous effect of postsynaptic FMRP deletion in L2/3 neurons in a primarily WT cortex on synapses from WT callosal axons. At P18-30, acute cortical slices were prepared and simultaneous whole cell recordings were performed on pairs of neighboring L2/3 pyramidal neurons in left barrel cortex with one being a GFP (-), or WT neuron, and the other a GFP (+), or *Fmr1* KO neuron (*Figure 2A–B*). Monosynaptic EPSCs from callosal axons were elicited with either bulk LED stimulation or blue laser, to perform sCRACM, as in *Figure 1* (*Figure 2C*, *Figure 2—figure supplement 2A-B*). Interestingly, at the earliest age for recording (P18-20), overall callosal synaptic input strengths were not different between WT and postsynaptic *Fmr1* KO neurons (*Figure 2D*). However, at P23-30, postsynaptic *Fmr1* KO neurons had a 40 % reduction in callosal synaptic input strength compared to neighboring WT neurons (*Figure 2D*). A two-way ANOVA indicates a significant interaction between genotype and age (*Fmr1* x Age ***p < 0.001, F(1, 32) = 13.87)(*Figure 2E*). In addition, we observed a strong trend of callosal input strength to increase with developmental age in WT, but not *Fmr1* KO, neurons (*Figure 2E*). Similar results were obtained with sCRACM which revealed a ~ 45 % reduction in callosal input strength onto proximal dendrites of *Fmr1* KO neurons (*Figure 2—figure supplement 2C-E*). These data demonstrate that *Fmr1* in postsynaptic L2/3 neurons cell-autonomously promotes the development and/or strengthening of callosal synaptic inputs. Weak callosal synaptic strength in *Fmr1* KO neurons at P23-30 could be a consequence of deficient or delayed callosal synapse maturation and may normalize in the adult. To test this possibility, we repeated experiments deleting postsynaptic *Fmr1* with a P1 AAV Cre-GFP injection and recorded LED-evoked EPSCs onto pairs of neighboring WT and *Fmr1* KO L2/3 neurons in adult mice (P57-65). Similar to young mice, callosal mediated EPSCs were weak in *Fmr1* KO neurons; reduced by 25%, in comparison to WT neurons (*Figure 2F–G*). These results indicate that weak callosal synaptic transmission persists into adulthood with postsynaptic *Fmr1* deletion.

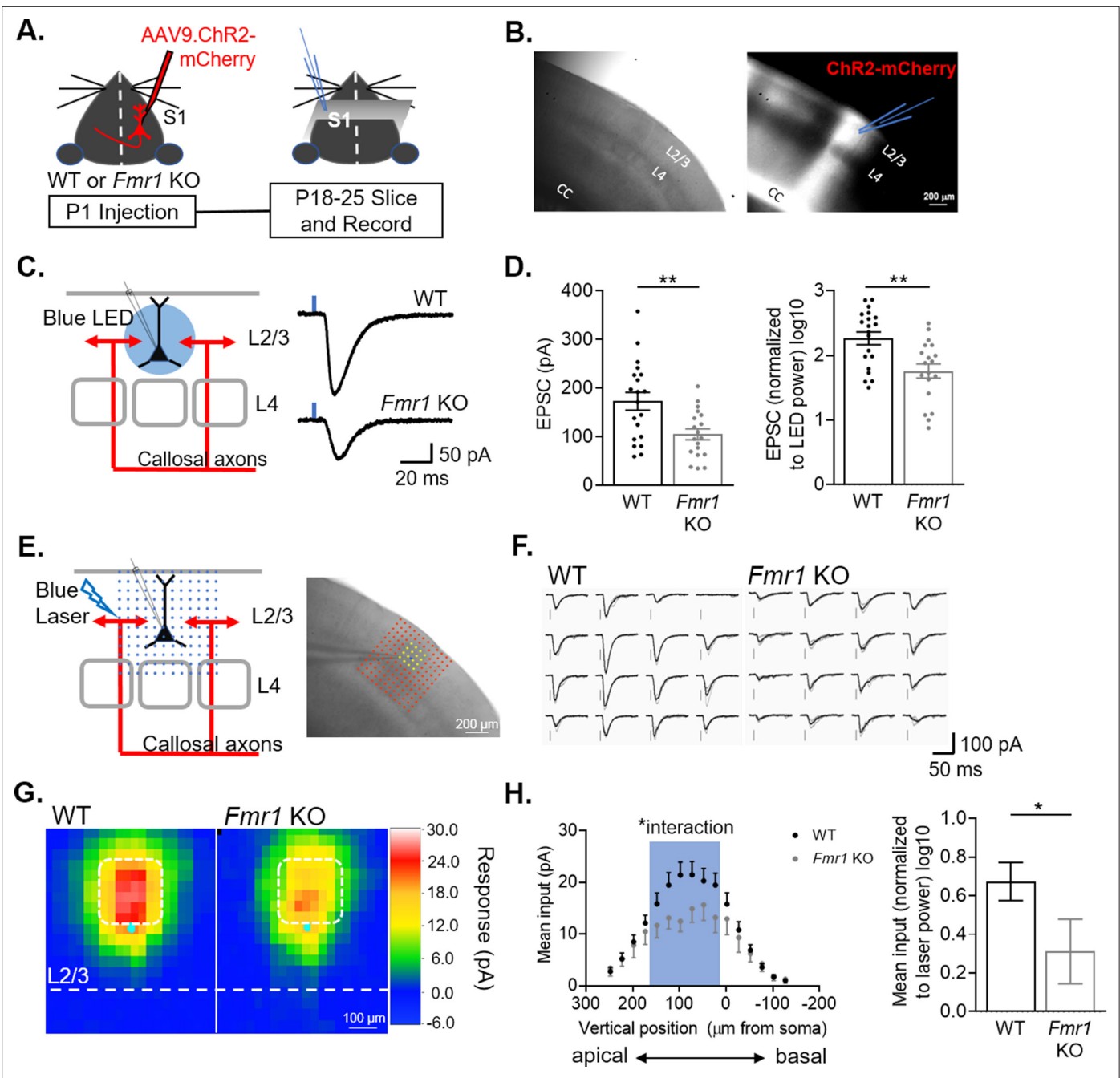

**Figure 1.** L2/3 pyramidal neurons in barrel cortex of *Fmr1* KO mice have weak callosal synaptic inputs. (**A**) Timeline and schematic of experimental paradigm. (**B**) Example image of S1 contralateral to AAV ChR2-mCherry injection. Left: DIC. Right: Red fluorescence of ChR2-mCherry labeled axons in corpus callosum (CC) and cortex. Recordings were performed on L2/3 neurons in an area of mCherry fluorescence. (**C**) LED stimulation paradigm (left) and example EPSCs from WT and *Fmr1* KO mice (right). Blue rectangle = 2 ms blue LED flash. (**D**) Left: Raw LED-evoked EPSC amplitudes in WT and *Fmr1* KO animals (WT = 172 ± 18 pA, n = 20; KO = 105 ± 11 pA, n = 19; unpaired t-test); Right: LED-evoked EPSC amplitudes normalized to LED power (WT = 2.2 ± 0.1, n = 20; KO = 1.7 ± 0.1, n = 19; unpaired t-test). (**E**) Schematic (left) and example experiment (right) of grid of blue laser stimulation during sCRACM relative to recorded neuron. (**F**) Example laser-evoked EPSCs at the locations highlighted in yellow in E from WT and *Fmr1* KO mice. (**G**) Group average of EPSC amplitudes evoked at different locations relative to the cell soma in WT and *Fmr1* KO mice. Individual maps are aligned by the location of soma (cyan dot). Pixel color represents the average amplitude of EPSCs evoked from that location. (**H**) Left: Vertical profile of mean synaptic input strength (mean input – average of EPSC amplitudes from individual locations within a specific area)(soma, x = 0) (*Fmr1* n.s. p = 0.16, F(1, 60) = 2.060; *Fmr1* x vertical position, F(15, 849) = 1.723; mixed-effects model). Right: Mean of EPSC amplitudes in the outlined area in G (white) and left graph (blue), normalized to laser power. (WT = 0.674 ± 0.098, n = 40; KO = 0.312 ± 0.167, n = 22; Mann Whitney). For this and all figures, error bars represent standard error mean (SEM). *p < 0.05, **p < 0.01.

*Figure 1 continued on next page*

*Figure 1 continued*

The online version of this article includes the following source data for figure 1:

**Source data 1.** LED induced EPSC amplitudes and quantification of sCRACM maps.

## Selective weakening of AMPAR-, but not NMDAR-, mediated synaptic transmission from callosal inputs onto *Fmr1* KO L2/3 pyramidal neurons

Weak callosal-mediated EPSCs onto postsynaptic *Fmr1* KO L2/3 pyramidal neurons could be due to reductions in the number of synaptic connections, presynaptic release probability, the strength of individual synapses, or any combination of these. To further investigate the synaptic basis of weak callosal inputs in P23-30 slices, we recorded in strontium ($Sr^{2+}$), substituted for $Ca^{2+}$, in the external ASCF which results in asynchronous vesicle release and measurement of quantal synaptic events evoked from callosal axons by the LED (*Oliet et al., 1996*). Changes in the amplitude of quantal events reflect synapse strength, whereas changes in the frequency of events reflect changes in synapse number and/or presynaptic release probability. Postsynaptic *Fmr1* KO neurons received >20% fewer evoked events in $Sr^{2+}$ in comparison to neighboring WT neurons with no change in event amplitude (*Figure 3A–C*). The baseline frequency and amplitude of spontaneous EPSCs, prior to LED stimulation, were not different between genotypes. These results suggest the reduced EPSCs onto postsynaptic *Fmr1* KO neurons are a consequence of reduced presynaptic release probability, functional synapse number or both. To further test this conclusion, we measured the coefficient of variance (C.V.) of LED-evoked callosal EPSCs onto WT and *Fmr1* KO neurons (from experiments in *Figure 2D*, *Figure 4* and Figure 5B). C.V. is inversely proportional to release probability and synapse number (*Manabe et al., 1993*) and was increased by 18 % in *Fmr1* KO neurons (*Figure 3—figure supplement 1*). This result together with reduced frequency of events evoked in $Sr^{2+}$ indicate that the weakening of callosal-mediated EPSCs in *Fmr1* KO is due in part to decreased synapse number and/or presynaptic release probability.

To further differentiate between release probability and functional synapse number, we measured NMDA receptor (R) mediated EPSCs from callosal axons onto WT or postsynaptic *Fmr1* KO L2/3 neurons. Because NMDARs are colocalized with AMPARs at excitatory synapses, reduced glutamate release probability or callosal synapse number onto *Fmr1* KO neurons would be expected to result in weak NMDAR-EPSCs. To evoke isolated NMDAR EPSCs from callosal axons, we included the AMPAR antagonist DNQX, glycine, and low $Mg^{2+}$ (0.1 mM) in the ACSF and voltage clamped neurons at –70 mV. In contrast to AMPAR-mediated EPSCs, amplitudes of NMDAR EPSCs evoked from callosal axons were not different between WT and neighboring postsynaptic *Fmr1* KO neurons (*Figure 3D*). This observation suggests that the NMDAR content within callosal synapses are similar for WT and KO neurons and that there is no change in presynaptic release probability or synapse number for the callosal axons. Taken together with the reduced frequency of AMPAR quantal events and increased C.V. (*Figure 3C*, *Figure 3—figure supplement 1*), our results suggest that L2/3 *Fmr1* KO neurons have an increased proportion of synapses with NMDARs, but not functional AMPARs, indicative of immature synapses.

## Postsynaptic *Fmr1* KO pyramidal neurons in both L2/3 and L5 receive weak callosal synaptic inputs with action-potential-driven synaptic transmission

For experiments described in *Figures 1 and 2*, we included TTX and 4-AP in the bath to isolate evoked EPSCs from ChR2-expressing callosal axons and block potential contamination from polysynaptic local L2/3 inputs. A caveat of this approach is that callosal synaptic transmission is not triggered by action potentials, but by direct depolarization and activation of voltage-gated $Ca^{2+}$ channels at the callosal axon terminal. To determine if similar results are observed with action potential-evoked synaptic transmission, we repeated experiments without TTX/4AP, but increased $Ca^{2+}/Mg^{2+}$ and added the NMDAR blocker, CPP, to reduce polysynaptic responses from local circuits. We also used a blue laser to depolarize a local (30 μm) area of callosal axons while limiting activation of local circuits (*Figure 5A*). Quantification of mean EPSC amplitude within the region covering the soma and major dendrites shows that L2/3 pyramidal neurons with postsynaptic *Fmr1* deletion have a 45 % reduction in callosal synaptic input strength as compared to neighboring WT neurons (*Figure 5A–B*).

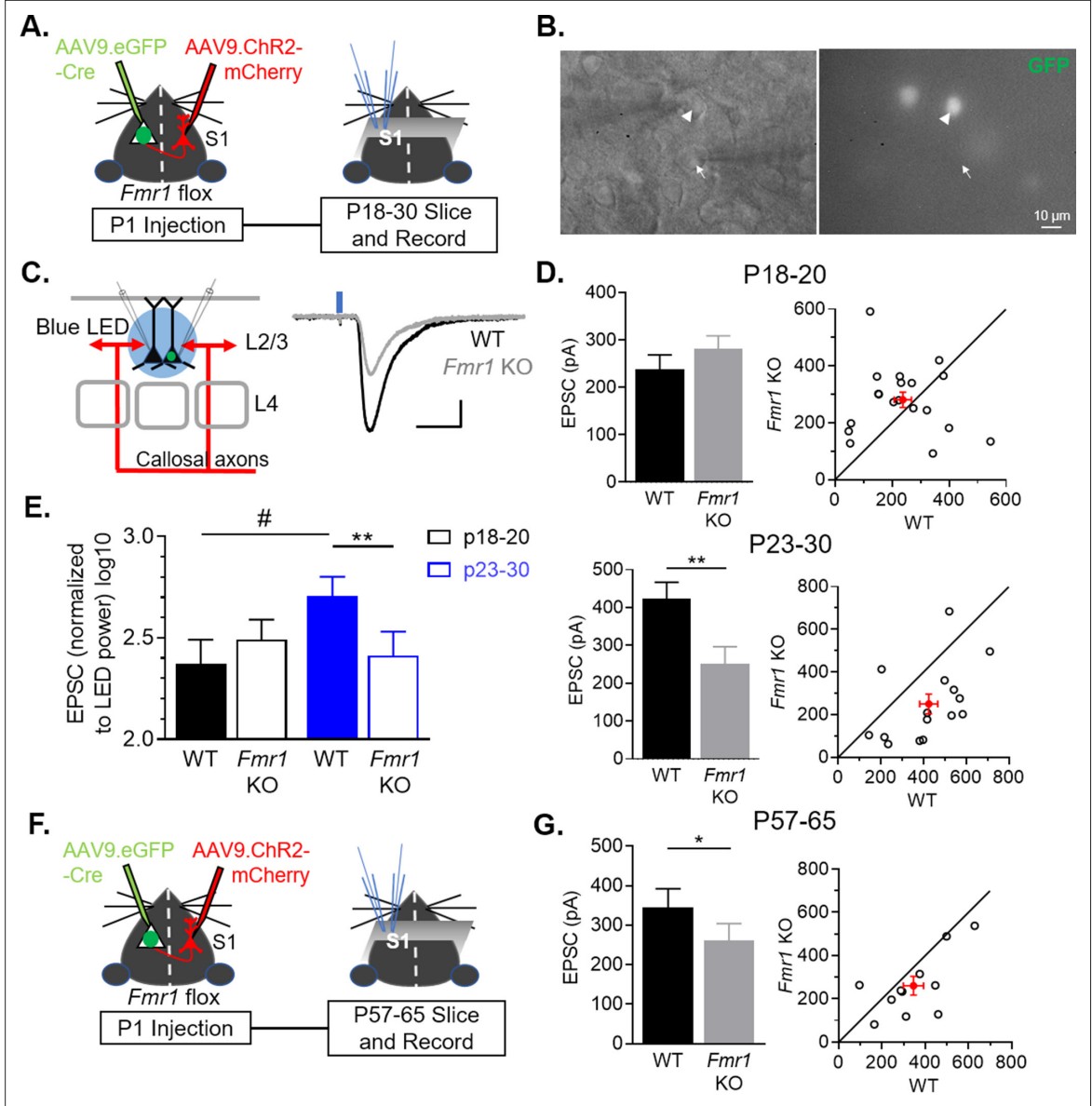

**Figure 2.** Postnatal and postsynaptic deletion of *Fmr1* in L2/3 pyramidal neurons cell autonomously weakens callosal synaptic inputs. (**A**) Timeline and schematic of experimental paradigm for juvenile recordings. (**B**) Simultaneous patch clamping of a neighboring AAV Cre-GFP+, *Fmr1* KO (arrowhead) and GFP-, *Fmr1* WT (arrow) pyramidal neurons in L2/3 of barrel cortex. Left: DIC. Right: Green fluorescence. (**C**) LED bulk stimulation paradigm (left) and example EPSCs from a pair of WT and *Fmr1* KO neurons (right). Scale = 100 pA, 20 ms. (**D**) Left: Group average of LED-induced EPSC amplitudes in WT and *Fmr1* KO neurons at P18-20 (top) (WT = 238 ± 30 pA; KO = 281 ± 27 pA, n = 19 pairs, n.s.) and P23-30 (bottom) (WT = 424 ± 42 pA; KO = 250 ± 46 pA, n = 15 pairs; paired t-test); Right: EPSC amplitudes from individual cell pairs (open circles). Mean ± SEM (filled circle). Diagonal line represents equality. (**E**) EPSC amplitudes, normalized to LED power, across different ages (replot from **D**) (*Fmr1* x Age ***p < 0.001, F(1, 32) = 13.87; *Fmr1*, Age, ns, ANOVA; #p < 0.1, multiple comparison). (**F**) Timeline and schematic of 2 month old recordings. (**G**) Left: Group average of LED-induced EPSC amplitudes in WT and *Fmr1* KO neurons at 2 month old (WT = 346 ± 47 pA; KO = 260 ± 43 pA, n = 11 pairs, paired t-test); Right: Distribution of values from individual cell pairs. *p < 0.05; **p < 0.01;***p < 0.001.

The online version of this article includes the following source data and figure supplement(s) for figure 2:

**Source data 1.** LED induced EPSC amplitudes from different age groups.

**Figure supplement 1.** Sparse transfection of Cre-GFP in the recorded hemisphere.

**Figure supplement 2.** Weakening of callosal synaptic inputs onto postsynaptic *Fmr1* KO L2/3 neurons is confirmed by sCRACM with spatial distribution.

**Figure supplement 2—source data 1.** Quantification of sCRACM maps.

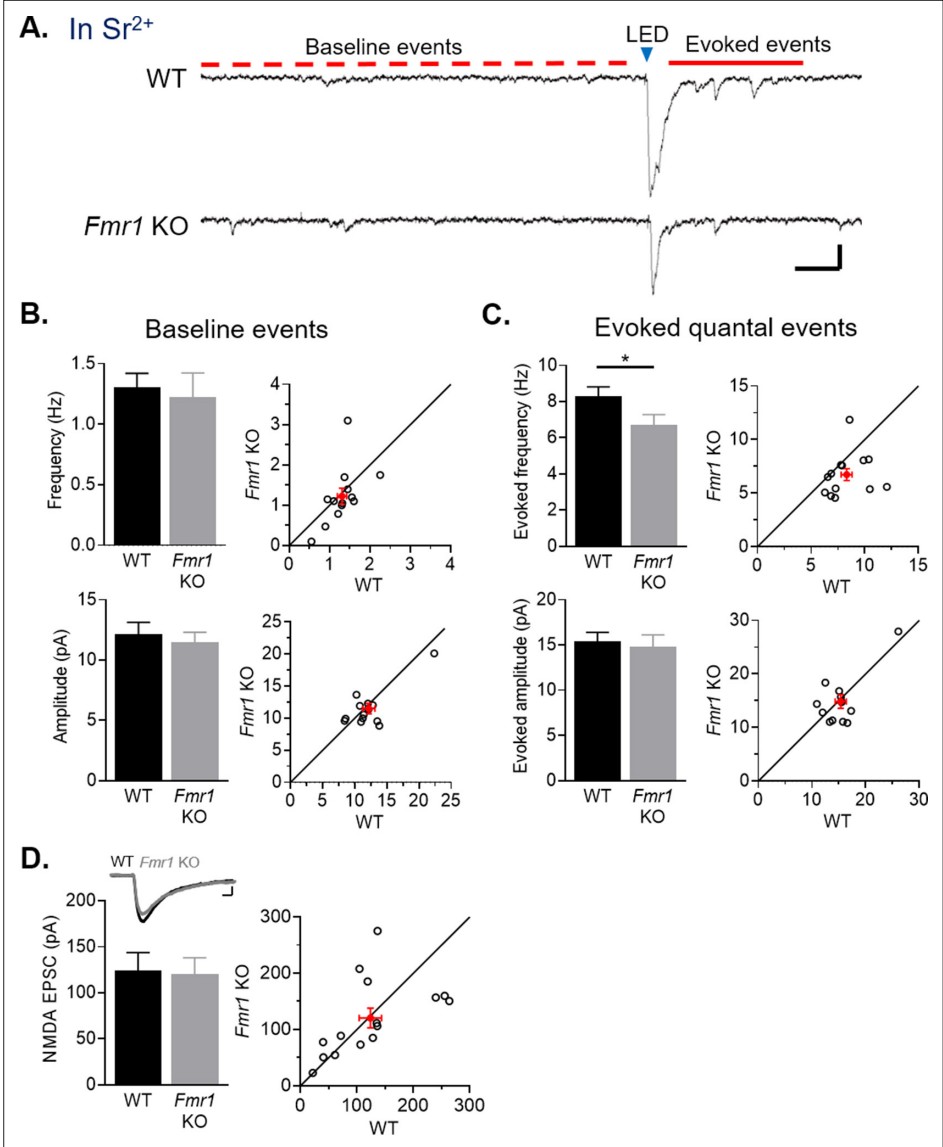

**Figure 3.** Callosal synapses onto *Fmr1* KO neurons have reduced quantal event frequency and a selective weakening of AMPA receptor-mediated synaptic transmission. (**A**) Example traces of Sr$^{2+}$-evoked quantal events. Scale = 25 pA, 100 ms. Baseline, spontaneous events are defined as those which occur within 1 s prior to LED flash (red dotted line) and evoked events occur 50–350 ms after LED flash (red line). (**B**) Left: Baseline frequency (top) (WT = 1.3 ± 0.1, KO = 1.2 ± 0.2 Hz, n.s., paired t-test) and amplitude (bottom) (WT = 12 ± 1, KO = 11 ± 1 pA, n.s., Wilcoxon test) of quantal EPSCs for WT and *Fmr1* KO neuron pairs; Right: distribution of values from individual cell pairs. (**C**) Left: Evoked frequency (top) (WT = 8.3 ± 0.5, KO = 6.7 ± 0.5 Hz, *p < 0.05, Wilcoxon test) and amplitude (bottom) (WT = 15 ± 1, KO = 15 ± 1 pA, n.s., Wilcoxon test, n = 13 pairs) of quantal EPSCs for WT and *Fmr1* KO neuron pairs; Right: distribution of values from individual cell pairs. (**D**) Top: Example NMDAR EPSCs from a WT and *Fmr1* KO pair. Scale = 25 pA, 20 ms. Bottom: LED-induced NMDAR EPSC amplitudes of WT and *Fmr1* KO neuron pairs (WT = 124 ± 20, KO = 120 ± 18 pA, n.s., paired t-test, n = 15 pairs); Right: distribution of values from individual cell pairs.

The online version of this article includes the following source data and figure supplement(s) for figure 3:

**Source data 1.** Quantal events and NMDAR EPSC.

**Figure supplement 1.** Coefficient of variance (C.V.) of LED-evoked EPSCs from P23-30 cell pairs in *Figure 2D* and 5B₁.

**Figure supplement 1—source data 1.** CV analysis.

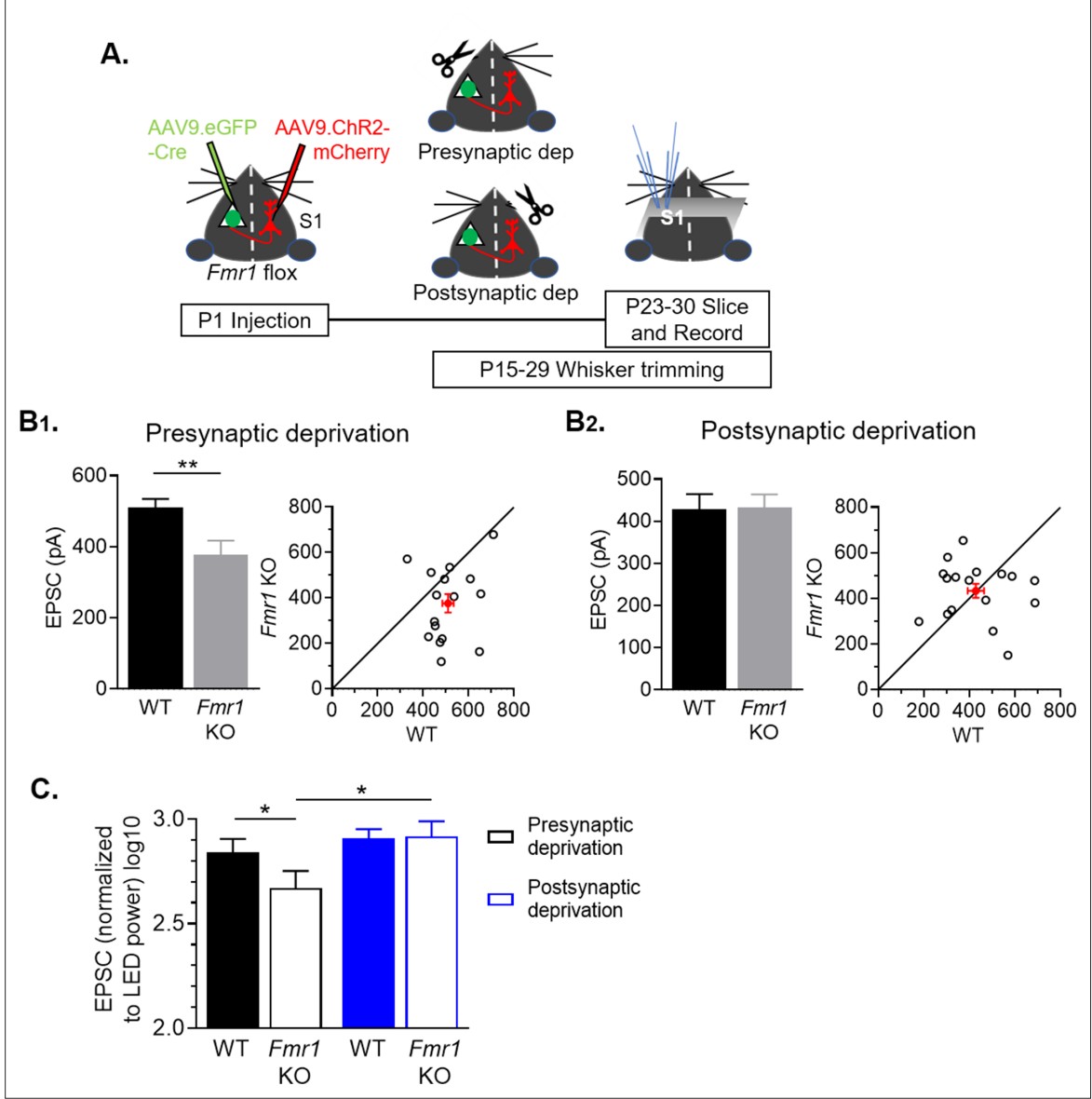

**Figure 4.** Sensory deprivation by whisker trimming normalizes callosal input strength onto postsynaptic *Fmr1* KO neurons. (**A**) Experimental paradigm. Trimming the whisker pad either ipsilateral or contralateral to the AAV-Cre-GFP injected hemisphere deprived either the presynaptic callosal projection neurons or postsynaptic *Fmr1* KO neurons, respectively, of patterned sensory experience- driven activity. (**B₁, B₂**) Left: Raw LED-induced EPSC amplitudes in WT and *Fmr1* KO neurons with presynaptic deprivation (WT = 510 ± 25; KO = 376 ± 41 pA, n = 16 pairs, **p < 0.01, paired t-test) or with postsynaptic deprivation (WT = 428 ± 36; KO = 433 ± 31 pA, n = 17 pairs, n.s., paired t-test): Right: Values from individual cell pairs. (**C**) LED-induced EPSC amplitudes normalized to LED power (replot from (**B**)) (*Fmr1* x deprivation interaction *p < 0.05, F(1, 31) = 4.977, ANOVA; presynaptic deprivation WT vs. *Fmr1* KO, *p < 0.05, *Fmr1* KO presynaptic deprivation vs. postsynaptic deprivation, *p < 0.05, Sidak's multiple comparisons).

The online version of this article includes the following source data and figure supplement(s) for figure 4:

**Source data 1.** LED induced EPSC amplitudes with deprivation.

**Figure supplement 1.** Sensory experience dependent weakening of callosal inputs is confirmed by sCRACM.

**Figure supplement 1—source data 1.** Quantification of mean inputs.

**Figure supplement 2.** Miniature (m) EPSC frequency and amplitude, as well as input resistance from pairs of WT and postsynaptic *Fmr1* KO L2/3 pyramidal neurons in P23-30 sensory intact, presynaptic sensory deprived and postsynaptic sensory deprived animals.

**Figure supplement 2—source data 1.** Miniature EPSC and input resistance.

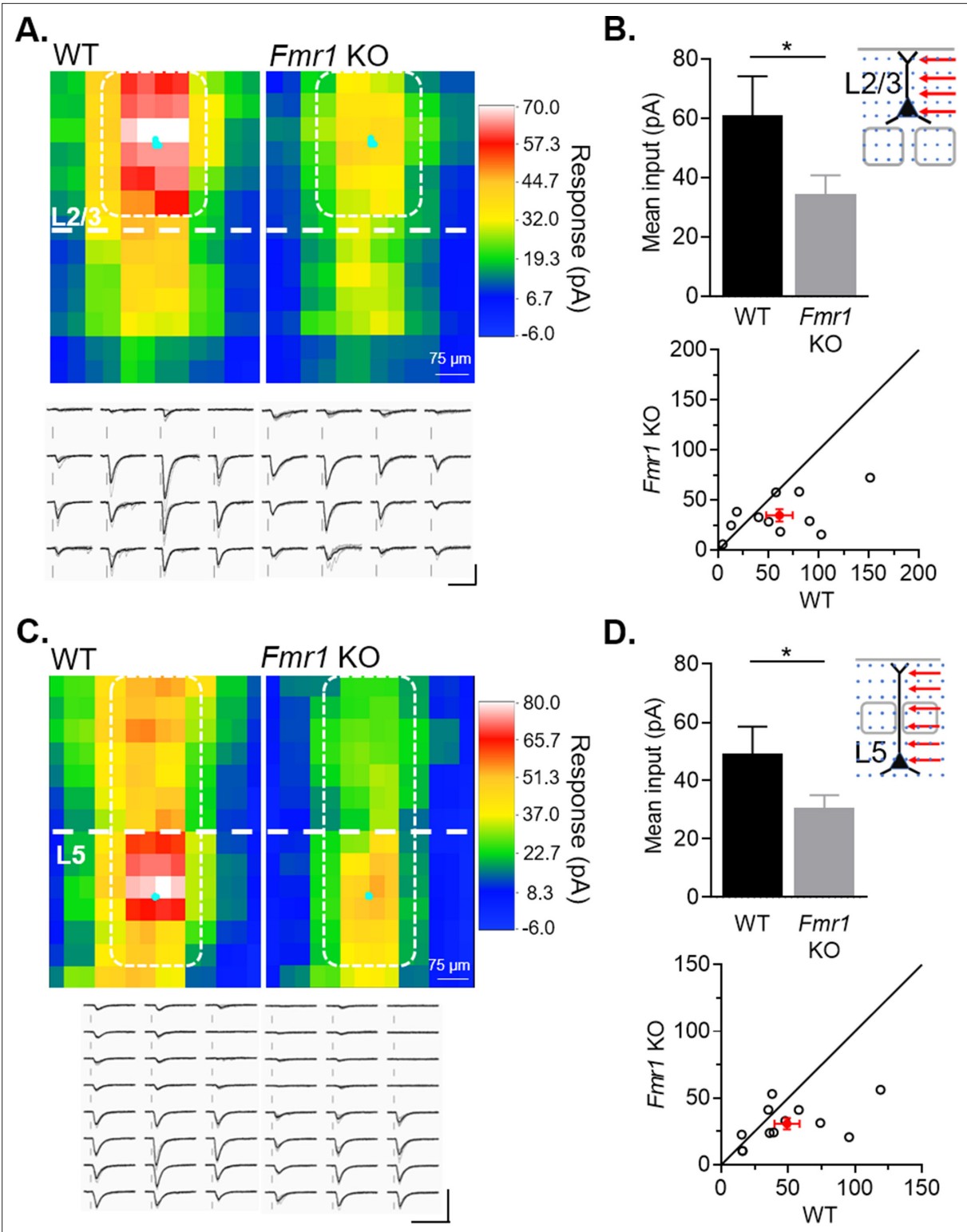

**Figure 5.** Cell autonomous deletion of *Fmr1* in postsynaptic L2/3 or L5 neurons results in weak action-potential driven synaptic transmission from callosal inputs. (**A**) Top: Average color map of action-potential-mediated callosal synaptic input strengths onto pairs of WT and *Fmr1* KO L2/3 pyramidal neurons at P23-30; Bottom: Example responses from the highlighted positions (white). Scale = 100 pA, 50 ms. (**B**) Top: Mean callosal inputs strength from area highlighted in white in A (WT = 61 ± 13, KO = 35 ± 6 pA, n = 11 pairs, *p < 0.05, paired t-test); Bottom: Distribution of values from individual cell pairs. (**C**) Top: Average color map of callosal input strengths onto pairs of WT and *Fmr1* KO L5 pyramidal neurons at P23-30; Bottom: Example responses from the highlighted area (white). Scale = 200 pA, 60 ms. (**D**) Top: Mean callosal input strengths onto L5 neurons from the area highlighted in

*Figure 5 continued on next page*

*Figure 5 continued*

white in C (WT = 49 ± 9, KO = 31 ± 4 pA, n = 12 pairs, *p < 0.05, paired t-test). Bottom: distribution of values from individual cell pairs.

The online version of this article includes the following source data for figure 5:

**Source data 1.** Quantification of mean inputs.

L5 pyramidal neurons are another major target of callosal axons (*Petreanu et al., 2007*; *Wang et al., 2007*). To determine if weak callosal synaptic inputs to postsynaptic *Fmr1* KO neurons are specific to L2/3, we performed the same experiment on L5 pyramidal neurons. Similar to L2/3, L5 pyramidal neurons with postsynaptic *Fmr1* deletion have a 40 % reduction in the strength of callosal inputs stimulated around the soma and apical dendrites (*Figure 5C–D*). Together, these results show that postsynaptic deletion of *Fmr1* in pyramidal cortical neurons generally weakens callosal synaptic inputs and this is observed with action potential driven synaptic transmission and across different layers.

## Sensory deprivation by whisker trimming normalizes callosal synaptic strengths in *Fmr1* KO L2/3 neurons

Targeting, branching and elaboration of callosal axons into the contralateral neocortex occurs post-natally (~P5-15) and depends on activity of presynaptic and postsynaptic cortical neurons as well as whisker sensory experience (*Huang et al., 2013*; *Mizuno et al., 2010*; *Wang et al., 2007*). This suggests that sensory experience may interact with FMRP to regulate development of callosal synapses. To test this idea, we sparsely deleted *Fmr1* in postsynaptic L2/3 neurons in the left barrel cortex and express ChR2-mCherry in callosal projecting neurons in the right hemisphere (as in *Figure 4A*). Beginning at P15, we unilaterally trimmed whiskers daily on the right whisker pad which would reduce the most direct ascending sensory-driven patterned activity to the left hemisphere containing L2/3 neurons with postsynaptic *Fmr1* deletion (postsynaptic deprivation condition). In littermates, we trimmed whiskers on the left whisker pad which would primarily deprive the L2/3 callosal projection neurons of sensory driven patterned activity (presynaptic deprivation condition) (*Figure 4A*). Whisker trimming began at P15 to reduce effects of sensory deprivation on the early growth and branching of callosal projection axons and continued until the day before slice recordings. Recordings of EPSCs evoked from callosal axons by either LED stimulation (*Figure 4B*) or sCRACM (*Figure 4—figure supplement 1*) were obtained from pairs of neighboring WT and *Fmr1* KO L2/3 neurons. Because we recorded from pairs of WT and postsynaptic *Fmr1* KO neuron neighbors, any effects of sensory deprivation on callosal axon innervation, growth, or branching in a given cortical region would be expected to similarly affect each genotype. Thus, our results reflect the cell autonomous effects of *Fmr1* on synaptic function or connectivity. In the 'presynaptic deprivation' condition, callosal synaptic input strength was weak (27 % reduction) onto postsynaptic *Fmr1* KO L2/3 neurons as compared to WT, similar to that observed in whisker intact mice (*Figure 2D*). In contrast, in the 'postsynaptic deprivation' condition, callosal synaptic inputs strengths were similar between WT and *Fmr1* KO neurons. To compare the callosal synaptic inputs strength within genotypes and across sensory deprivation paradigms, we normalized each LED induced EPSC to its stimulation power (*Figure 4C*). A two-way ANOVA revealed an interaction of deprivation condition and *Fmr1* (*p < 0.05, $F_{(1, 31)}$ = 4.977, ANOVA). Surprisingly, callosal synaptic input strengths in WT neurons were not different between deprivation conditions. In contrast, in *Fmr1* KO neurons callosal input strengths were weaker, decreased by ~45%, in the presynaptic deprivation condition, as compared to postsynaptic deprivation. These data suggest that whisker experience-driven activity of postsynaptic *Fmr1* KO L2/3 neurons weakens callosal synaptic inputs.

## Input specific strengthening of callosal synaptic connections by postsynaptic FMRP

A previous study reported weak L4 to L2/3 synaptic inputs in the barrel cortex of global *Fmr1* KO mice (*Bureau et al., 2008*), suggesting that *Fmr1* generally promotes excitatory synapse strength onto L2/3 neurons regardless of whether they are from local or long-range sources. To determine if postsynaptic *Fmr1* promotes excitatory synapse development from local cortical circuits in a cell autonomous manner as it does for callosal inputs, we assessed local input strengths using Laser Scanning

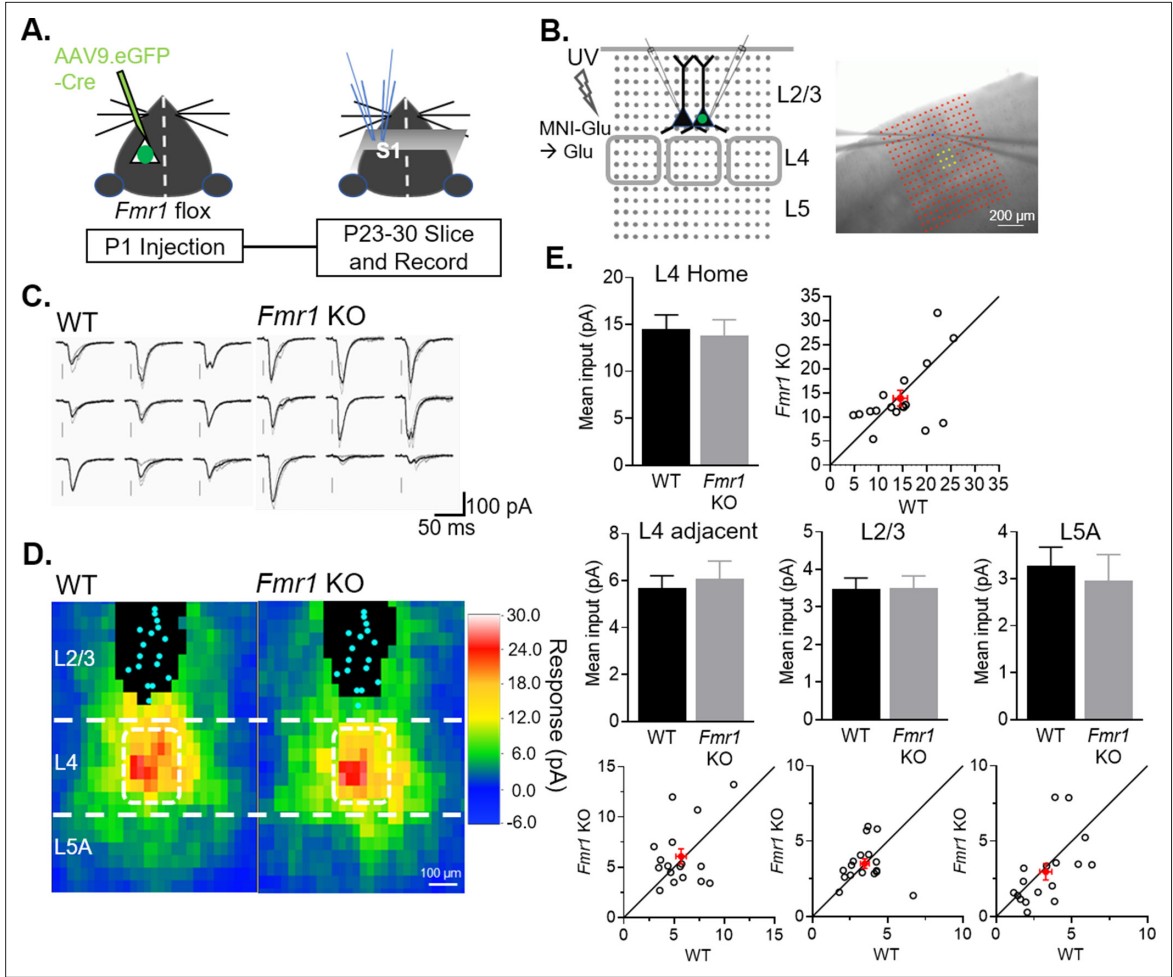

**Figure 6.** Postsynaptic deletion of *Fmr1* in L2/3 neurons does not affect excitatory synaptic inputs from local columnar circuit. (**A**) Experimental design schematic. (**B**) Left: Position of UV laser scanning photostimulation (LSPS) grid (grey dots) relative to cortical layers and recorded neurons in L2/3 and L4 barrels (gray squares). Right: IR-DIC image of dual recordings in L2/3 in a slice with laser stimulation grid (red). Yellow dots indicate L4 home barrel. (**C**) Example of EPSCs in a pair of WT and *Fmr1* KO L2/3 neurons in response to LSPS and glutamate uncaging in L4 home barrel (yellow dots in B; right). (**D**) Color map of spatial distribution of average synaptic input strengths in response to LSPS. L4 home barrel (white rectangle). Cyan dots represent locations of soma and black pixels are direct responses. (**E**) Mean synaptic input strength from L4 home barrel, L4 adjacent barrels, adjacent L2/3 and L5A (WT vs. *Fmr1* KO, n.s., n = 17 pairs, Wilcoxon or paired t-test) and values from individual cell pairs.

The online version of this article includes the following source data and figure supplement(s) for figure 6:

**Source data 1.** Mean inputs from LSPS maps.

**Figure supplement 1.** Postsynaptic deletion of *Fmr1* does not affect local columnar circuit inputs onto L2/3 pyramidal neurons at 2 weeks of age.

**Figure supplement 1—source data 1.** L4 mean inputs from LSPS maps.

**Figure supplement 2.** Summary model of the role of postsynaptic *Fmr1* on development of callosal and local neocortical synapses.

Photo-Stimulation (LSPS) with glutamate uncaging. Slices were bathed in MNI-caged glutamate and pseudorandom flashes of a UV laser (355 nm) beam at individual locations (20 μm diameter) within a 16-by-16 grid surrounding the recorded L2/3 neurons focally released glutamate to evoke action potentials and synaptic transmission from neurons at that location. The grid for UV laser flashing was positioned to stimulate neurons in Layers 2/3–5 of home and adjacent barrels. At P23-30, LSPS was performed on pairs of simultaneously recorded WT and neighboring L2/3 neurons with postsynaptic *Fmr1* deletion (*Figure 6A–B*). The amplitude of monosynaptic EPSCs evoked from each position in the slice for an individual neuron were converted into a color map and then individual maps were aligned to the home barrel to create an average color map per genotype (*Figure 6C–D*). Responses from direct glutamate activation onto recorded neurons, as described in methods, were excluded

from analysis, and represented by black pixels. EPSCs evoked from L4 home or adjacent barrels, L5A or L2/3 were unaffected on *Fmr1* KO L2/3 neurons (**Figure 6E**), in contrast to the weak EPSCs from callosal inputs.

Bureau and colleagues demonstrated weak L4 to L2/3 synaptic strength in the global *Fmr1* KO when measured at 2 weeks of age, but the difference was diminished at 3 weeks, suggestive of a developmental delay. To test if loss of postsynaptic *Fmr1* weakens local L4 to L2/3 synaptic inputs at early developmental stages, we performed LSPS on pairs of WT and postsynaptic *Fmr1* KO L2/3 pyramidal neurons at P14-17. Similar to results obtained at P23-30, we observed normal L4 and adjacent L2/3 synaptic input strengths onto postsynaptic *Fmr1* L2/3 neurons recorded at P14-17 (**Figure 6— figure supplement 1A**). We also tested if embryonic deletion of postsynaptic *Fmr1* was necessary to affect L4 to L2/3 synapse development, using slices from females with heterozygous (het) and mosaic expression of *Fmr1*, as we have described (**Patel et al., 2013**; **Patel et al., 2014**). Briefly, X-linked GFP mice were bred with *Fmr1* KO males. Due to random X-chromosome inactivation in the embryo, female *Fmr1* het mice offspring have a mosaic expression of GFP(+) WT and GFP(-) *Fmr1* KO neurons. LSPS maps were performed on pairs of WT and *Fmr1* KO neurons at P14-17 where we observed normal L4 input strengths onto L2/3 *Fmr1* KO neurons (**Figure 6—figure supplement 1B**). In contrast to results with postsynaptic *Fmr1* deletion, we observed weak L4-to-L2/3 synaptic strengths with LSPS in the global *Fmr1* KO (**Figure 6—figure supplement 1C**) similar to that reported by **Bureau et al., 2008**. However, unlike Bureau et al, we observed weak L4 inputs at later ages (P18-25; **Figure 6— figure supplement 1C**). Taken together our results implicate FMRP in postsynaptic L2/3 neurons in the input specific development of callosal synapses. Although synaptic inputs from L4 to L2/3 are weak in the global *Fmr1* KO, this is not a postsynaptic, cell autonomous function of FMRP, but instead is either a non-cell autonomous function of FMRP or a role in presynaptic L4 neurons.

## Discussion

Reduced interhemispheric connectivity, observed both structurally and functionally, is a hallmark of ASD in humans and correlated with symptoms (**Dimond et al., 2019**; **Holiga et al., 2019**; **Li et al., 2019**; **O'Reilly et al., 2017**; **Rane et al., 2015**; **Yao et al., 2021**). However, little is known of the cellular, synaptic, and molecular mechanisms by which this occurs in ASD and any direct role of ASD-risk genes. Here, we demonstrate a direct, postsynaptic and postnatal role for *Fmr1* in maturation and/or stability of callosal synaptic inputs in L2/3 and L5 cortical neurons and this change is also observed in the FXS- mouse model, the global *Fmr1* KO. Surprisingly, postsynaptic deletion of *Fmr1* did not weaken synaptic inputs from local columnar circuits onto L2/3, revealing that postsynaptic FMRP differentially regulates development of select synaptic inputs (**Figure 6—figure supplement 2**). Sensory deprivation of postsynaptic *Fmr1* KO neurons prevented weakening of callosal synaptic inputs suggesting that experience-driven patterned activity of postsynaptic L2/3 neurons without FMRP is necessary for synaptic weakening and/or prevents maturation. In conclusion, our results reveal a postsynaptic mechanism by which *Fmr1* regulates callosal connectivity that likely contributes to the reduced interhemispheric structural and functional connectivity in *Fmr1* KO mice and humans with FXS (**Haberl et al., 2015**; **Swanson et al., 2018**; **Zerbi et al., 2018**).

### A synaptic basis for reduced functional long-range connectivity in FXS

Functional MRI studies show that *Fmr1* KO mice have reduced corticocortical long-range connectivity, especially among the sensory and motor cortices (**Haberl et al., 2015**; **Zerbi et al., 2018**) as well as reduced corpus callosum structural integrity. By investigating callosal synaptic connections between bilateral barrel cortices, we find weak functional synaptic inputs at these long-range connections in L2/3 in *Fmr1* KO mice (**Figure 1**). This result suggests that reduced functional coherence between bilateral barrel cortices in *Fmr1* KO mice could be due to a disrupted communication between the cortices through the weakened monosynaptic transmission. Interhemispheric functional connectivity is thought to shape perceptual integration including those involved in speech comprehension and global form processing, domains that are impaired in ASD (**Booth and Happé, 2018**; **Friederici et al., 2007**; **Happé and Frith, 2006**; **Peiker et al., 2015**; **Preisig et al., 2021**; **Simon and Wallace, 2016**). Based on fMRI results in the *Fmr1* KO (**Haberl et al., 2015**; **Zerbi et al., 2018**), it is likely that FMRP

promotes other 'long-range' synaptic connections between cortical areas, such as ipsilateral connectivity between S1 and M1/M2, as well as cortical-subcortical structures.

## Postsynaptic FMRP in L2/3 neurons promotes maturation of callosal synapses

Using in vivo sparse deletion and a simultaneous recording paradigm, we can stimulate the same set of callosal axons for a pair of WT and *Fmr1* KO neurons and directly compare their synaptic inputs. Our observation of weaker callosal synaptic inputs onto cell autonomous *Fmr1* KO L2/3 neurons (*Figure 2*) confirms results in *Fmr1* global KO mice and further reveals an essential role of postsynaptic FMRP in promoting callosal synapses development. With postsynaptic *Fmr1* KO we observe weak callosal synapses at P23-30, but not at P18-20. To determine if a similar developmental profile is observed in the global *Fmr1* KO, we analyzed a subgroup of data collected at P18-20 and observe weak callosal synaptic strength in the *Fmr1* KO at this early time point (WT = 181 ± 22 pA, n = 15; KO = 106.1 ± 17.77 pA, n = 7, *$P < 0.05$, unpaired t-test). The later developmental onset of weak callosal synaptic transmission with postsynaptic *Fmr1* deletion may be due to later postnatal deletion of *Fmr1* using P1 AAV-Cre-GFP injection which we estimate to occur about P7-P9 (*Rajkovich et al., 2017*). However, we cannot rule out a role for FMRP in presynaptic, callosal projecting neurons, or other cell types, such as oligodendrocytes, within the first postnatal weeks to establish callosal synaptic connections (*Doll et al., 2020*; *Hanson and Madison, 2007*; *Patel et al., 2013*).

Our results indicate that callosal synapses do not mature or are not maintained without FMRP in postsynaptic L2/3 neurons. In support of a role in maturation, callosal synaptic strength tends to increase from P18-20 to P23-30 in WT neurons but not in postsynaptic *Fmr1* KO neurons (*Figure 2E*). Furthermore, we observe a selective weakening of AMPAR, but not NMDAR-, mediated synaptic transmission at callosal inputs onto *Fmr1* KO neurons (*Figure 3*). This result, together with a decreased frequency of evoked quantal events in $Sr^{2+}$ and increased coefficient of variation, suggests a reduced number of mature synapses with functional AMPARs. As synapses mature, they acquire NMDARs prior to AMPARs and NMDAR-only, immature synapses are often termed 'silent' synapses (*Ashby and Isaac, 2011*; *Hanse et al., 2013*). Although we did not measure 'silent' callosal synapses here, our findings would predict more 'silent' and immature callosal synapses in *Fmr1* KO L2/3 neurons. In support of this idea, thalamocortical inputs to L4, another 'long-range' synaptic pathway, are delayed in their development in the *Fmr1* KO, as measured by acquisition of AMPARs (*Harlow et al., 2010*). Another possible explanation for our results is the optogenetic stimulation paradigm we used to evoke glutamate release from callosal axons saturates synaptic NMDARs, but not AMPARs, due to the higher affinity of glutamate for NMDARs (*Patneau and Mayer, 1990*). Additional experiments and/or methods are needed to confirm a selective decrease in AMPARs at *Fmr1* KO callosal synapses. The observations of reduced corpus callosum structural integrity and interhemispheric coherence with fMRI in adult *Fmr1* KO suggest weak callosal connectivity in adults (*Haberl et al., 2015*; *Zerbi et al., 2018*). Weak callosal synapses persist in adults with postsynaptic *Fmr1* deletion, suggesting a deficit in maturation or AMPAR insertion/stability, as opposed to developmental delay (*Figure 2F–G*). Reduced callosal axon diameter is observed in adult *Fmr1* KO which may be a consequence of weak or immature callosal synapses and contribute to reduced interhemispheric coherence in FXS (*Haberl et al., 2015*).

## Postsynaptic FMRP differentially regulates synaptic inputs from local and long-range cortical circuits

Neocortical pyramidal neurons integrate excitatory synaptic inputs from local and long-range circuits, including ipsilateral and contralateral cortical areas (*Feldmeyer, 2012*; *Gerfen et al., 2018*). An imbalance in local and long-range connectivity has been hypothesized to contribute to ASD in humans; specifically, hyperconnectivity of local circuits and underconnectivity of long-range circuits or between brain regions (*Belmonte et al., 2004*; *Courchesne and Pierce, 2005*; *O'Reilly et al., 2017*; *Rane et al., 2015*). If or how ASD genes regulate the balance of local and long-range synaptic connectivity is unknown. Here we demonstrate that postsynaptic FMRP differentially regulates development and/or maintenance of synaptic inputs from local and long-range cortical sources. L2/3 neurons with postsynaptic deletion of *Fmr1* have weak and immature callosal synaptic inputs but normal synaptic inputs from L4, adjacent L2/3 and L5 (*Figure 6*). Bureau et al. (*Bureau et al., 2008*) and we (*Figure 6—figure*

*supplement 1C*) find that L4 to L2/3 synapses are weak in the global *Fmr1* KO, but we do not observe this with embryonic or postnatal cell autonomous deletion of *Fmr1* in L2/3 neurons. Together these results suggest a role for FMRP in presynaptic L4 neurons in synapse development onto L2/3 neurons, which is consistent with the reported deficits in L4 axon morphology in the global *Fmr1* KO (*Bureau et al., 2008*).

Results in L5 also indicate differential regulation of local and long-range cortical connectivity by postsynaptic FMRP. Our previous work using multiple simultaneous recordings of locally connected L5A neurons, revealed hyperconnectivity of *Fmr1* KO L5 neurons with their immediate neighbors ( < 40 μm apart) in S1 at 4 weeks of age (*Patel et al., 2014*). Hyperconnectivity of L5 local subnetworks resulted from deficient developmental pruning between *Fmr1* KO L5 neurons and was observed in both the global *Fmr1* KO and with postsynaptic *Fmr1* deletion. Prefrontal L5 cortical neurons in *Fmr1* KO mice are similarly hyperconnected (*Testa-Silva et al., 2012*). In contrast to local hyperconnectivity, here we find that postsynaptic deletion of *Fmr1* in L5 neurons results in weak callosal synaptic inputs. Thus, *Fmr1* KO L5 pyramidal neurons are hyperconnected locally and under-connected to contralateral cortex; an effect that is mediated by cell-autonomous and postsynaptic deletion of FMRP. Such an effect may promote the reported imbalances in local and long-range functional connectivity observed in ASD individuals. Postsynaptic deletion of FMRP in L2/3 neurons did not affect synaptic inputs from other layers or between columns within L2/3 (*Figure 6*) suggesting that FMRP may selectively promote pruning of local connections in L5 neurons. Alternatively, FMRP may prune synaptic connections within very local cortical subnetworks (within 40 μm) in both L2/3 and L5, but not connections between layers and columns.

The molecular mechanisms by which postsynaptic FMRP differentially regulates L4 and callosal inputs to L2/3 neurons is unclear. L4 neurons synapse primarily on basal dendrites of L2/3 neurons, whereas callosal inputs are primarily on apical dendrites (*Bosman et al., 2011*; *Figure 1G*). Therefore, localized expression and translational regulation of specific dendritic mRNAs by FMRP at either basal or apical dendritic compartments could differentially affect L4 and callosal inputs. Alternatively, FMRP translational regulation of postsynaptic cell adhesion molecules, such as neuroligins, could differentially impact specific presynaptic inputs based on their expression of binding partners such as neurexin splice variants (*Südhof, 2017*).

## Bidirectional regulation of callosal synaptic function by MEF2C and FMRP

Differential regulation of local and long-range cortical synaptic connectivity is observed with another ASD-risk gene, Myocyte Enhancer Factor 2 C (*Mef2c*). Loss-of-function mutations in *MEF2C*, which encodes an activity-dependent transcription factor, are implicated in intellectual disability, ASD and schizophrenia (*Assali et al., 2019*; *Mitchell et al., 2018*; *Rocha et al., 2016*). In contrast to FMRP, postsynaptic deletion of *Mef2c* in L2/3 neurons results in fewer and weak inputs from local circuits (L4, L2/3, and L5), but strengthened callosal inputs (*Rajkovich et al., 2017*). The differential regulation of local and callosal synaptic connections by postsynaptic MEF2C and FMRP, albeit in different directions, suggest that imbalances in local and long-range synaptic connectivity may contribute to different genetic causes of neurodevelopmental disorders. Our results also implicate roles for transcription and translational control in the input-specific development of cortical circuits. Bidirectional regulation of local and long-range connectivity by *Mef2c* in L2/3 or by *Fmr1* in L5 could be an effect of homeostasis or competition between local and long-range synaptic connections to maintain optimal cortical circuit function. For example, weakening of callosal synaptic input in *Fmr1* KO neurons may be compensatory and an attempt to normalize hyperconnected or hyperexcitable local circuits.

## Interaction of sensory experience and ASD-risk genes in regulation of long-range cortical circuits

The regulation of local or callosal connectivity by FMRP and MEF2C requires normal sensory experience suggesting that both of these genes function in experience and activity-regulated pathways necessary for cortical circuit development. Sensory deprivation by whisker trimming normalizes callosal synaptic inputs in L2/3 neurons with postsynaptic deletion of *Fmr1* (*Figure 4*) or *Mef2c* (*Rajkovich et al., 2017*). Specifically, trimming whiskers contralateral to L2/3 neurons with postsynaptic *Fmr1* deletion (postsynaptic deprivation) prevented callosal synaptic weakening. In contrast, 'presynaptic deprivation'

or trimming whiskers contralateral to ChR2-expressing, callosal projecting neurons, had no effect (*Figure 4*). This result suggests that sensory-driven patterned activity of postsynaptic *Fmr1* KO L2/3 neurons weakens callosal synapses. An intriguing possibility is that an activity-dependent long-term synaptic depression (LTD) process is enhanced at *Fmr1* KO L2/3 neurons as observed in hippocampal CA1 (*Huber et al., 2002*). Sensory deprivation also induces homeostatic synaptic scaling in primary sensory cortices, as measured with spontaneous or miniature (m) EPSCs (*Feldman, 2009*; *Hooks and Chen, 2020*). Neither the frequency nor amplitude of mEPSCs was different between neighboring WT and *Fmr1* KO L2/3 neurons in mice with normal sensory experience or 'presynaptic deprivation'. However, with 'postsynaptic deprivation', mEPSC amplitude and frequency was increased in *Fmr1* KO neurons relative to WT (*Figure 4—figure supplement 2*) suggestive of homeostatic up scaling in *Fmr1* KO neurons that may contribute to callosal input strengthening in this condition.

Bilateral underconnectivity is common in ASD as well as disconnection of other long-range cortical connections with other brain regions such as hippocampus and cerebellum. Bilateral connectivity and synchrony between cortical regions are necessary for speech comprehension, sensory processing, and cognition (*Bland et al., 2020*; *Castro et al., 2014*; *Friederici et al., 2007*; *Fries, 2009*; *Fries, 2015*; *Panzica et al., 2019*), domains impaired in FXS and ASD. Our present findings contribute to the understanding of the cellular and synaptic mechanisms by which ASD-risk genes, such as *FMR1*, regulate long-range connectivity, how this is coregulated and balanced with local circuit connectivity and interacts with experience-dependent brain development. Such information may contribute to therapies to aid abnormal brain connectivity in ASD.

# Materials and methods

**Key resources table**

| Reagent type (species) or resource | Designation | Source or reference | Identifiers | Additional information |
|---|---|---|---|---|
| Genetic reagent (*M. musculus*) | *Fmr1⁻/y* (male) | Jackson Laboratory | 003025 | |
| Genetic reagent (*M. musculus*) | *Fmr1ᶠˡ/ᶠˡ* (female) *Fmr1ᶠˡ/y* (male) | PMID:16257225 | | Dr. David Nelson (Baylor College of Medicine) |
| Genetic reagent (*M. musculus*) | X-linked GFP | Jackson Laboratory | 003116 | |
| Strain, strain background (*AAV*) | AAV9.CMV.HI.eGFP-Cre. WPRE.SV40 | Addgene | 105545 | |
| Strain, strain background (*AAV*) | AAV9.CAG.hChR2(H134R)-mCherry.WPRE. SV40 | Addgene | 100054 | |
| Chemical compound, drug | MNI-caged-L-glutamate | Tocris / HelloBio | 1490/ HB0423 | |
| Chemical compound, drug | (RS)-CPP | Tocris / HelloBio | 0173/ HB0036 | |
| Chemical compound, drug | 4-Aminopyridine (4-AP) | Sigma- Aldrich | A78403 | |
| Chemical compound, drug | DNQX disodium salt | Tocris | 2,312 | |
| Software, algorithm | LabView | National Instruments | RRID:SCR_014325 | |
| Software, algorithm | Multiclamp 700 A | Molecular Devices | RRID:SCR_021040 | |
| Software, algorithm | Prism 8 | Graphpad Software | RRID:SCR_002798 | |

## Animals

Fmr1 KO (*Fmr1⁻/y*) and X-linked GFP mice were obtained from Jackson laboratories (Stock No: 003025 and 003116, respectively). *Fmr1ᶠˡ/ᶠˡ* were obtained from Dr. David Nelson (Baylor College of Medicine) (*Mientjes et al., 2006*). Mice were maintained on a C57BL/6 J background and reared on a 12 hr light-dark cycle with access to food and water ad libitum. Male *Fmr1* WT (*Fmr1⁺/y*) with *Fmr1* KO (*Fmr1⁻/y*) littermates were used for experiments and both male and female pups were used for *Fmr1* flox (*Fmr1ᶠˡ/y* and *Fmr1ᶠˡ/ᶠˡ*). All animal experiments were conducted in accordance with the Institutional Animal Care and Use Committee (IACUC) at University of Texas Southwestern Medical Center.

## Viral transfections in neonatal mice

Commercially made AAV9.CMV.HI.eGFP-Cre.WPRE.SV40 (Addgene #105545) and AAV9.CAG. hChR2(H134R)-mCherry.WPRE.SV40 (Addgene #100054) were diluted to a titer ~$10^{12}$ vg/mL using sterile saline. Traces of Fast Green FCF dye (Sigma) were added to facilitate visualization of virus spreading. Neonatal mouse pups (P1) were first anesthetized by hypothermia, then fixed on a customized mold and placed on a stereotaxic frame. AAV9.eGFP-Cre (420–560 nL) was delivered to left ventricle at a depth of approximately 1.1 mm underneath skull through a beveled glass pipette using Nanoject II injector (Drummond Scientific, Inc). AAV9.ChR2-mCherry (400 nL) was delivered to superficial layers (0.5 mm underneath skull) of somatosensory cortex in right hemisphere at a speed of 1.2 µL/min using syringe pump (Harvard Apparatus, Inc).

## Acute slice preparation

Mice were anesthetized by intraperitoneal (i.p.) injection of Ketamine/Xylazine mixture and decapitated upon irresponsiveness to toe-pinch. Acute coronal slices (300 µm thickness) containing somatosensory barrel cortex were prepared using vibrating microtome (Leica VT1200S). During sectioning, tissue blocks were submerged in ice-cold dissection buffer containing (in mM): 110 choline chloride, 25 $NaHCO_3$, 25 dextrose, 11.6 ascorbic acid, 2.5 KCl, 1.25 $NaH_2PO_4$, 3.1 Na-pyruvate, 7 $MgCl_2$, and 0.5 $CaCl_2$, continuously aerated with 95%$CO_2$/5%$O_2$. For mice older than P23, transcardial perfusion of ice-cold dissection buffer was performed prior to decapitation to increase slice quality. Slices were then transferred to artificial cerebrospinal fluid (ACSF) solution containing (in mM): 125 NaCl, 25 $NaHCO_3$, 10 dextrose, 2.5 KCl, 1.25 $NaH_2PO_4$, 2 $MgCl_2$, and 2 $CaCl_2$ (aerated with 95%$CO_2$/5%O2) and recovered at 34 °C for 30 min followed by 30 min at room temperature.

## Electrophysiology

After recovery, slices were transferred to a recording chamber at room temperature and perfused with ACSF aerated with 95%$CO_2$/5%$O_2$. Slices were visualized by infrared differential interference contrast (IR-DIC) optics (Olympus BX51W1). Whole cell recordings of L2/3 and L5 pyramidal neurons were obtained using borosilicate pipettes (4–7 MΩ) and a Multiclamp 700 A amplifier (Molecular Devices). Internal solution contained (in mM): 130 K-gluconate, 10 HEPES, 6 KCl, 3 NaCl, 0.2 EGTA, 14 phosphocreatine-tris, 4 Mg-ATP and 0.4 Na-GTP. All recordings were conducted in voltage clamp, holding at –70 mV unless otherwise specified, and data were collected and analyzed using custom Labview programs (Labview 8.6, National Instruments Inc). Spiking patterns upon current injection were used as criteria to identify excitatory neurons. For experiments where spiking was blocked by TTX, rise time to hyperpolarizing current ( > 50 ms) and kinetics of mPSC (width at half height >2 ms) were used as criteria (*Povysheva et al., 2006*). Excitatory neurons with resting membrane potential < –50 mV and a series resistance <35 MΩ were included in analysis. Voltages were not corrected for junction potential. For simultaneous patch clamp recordings, the distance between the pair of cells (center-to-center) is 10–40 µm.

## Optogenetic bulk stimulation of callosal axons

Mice with either cortical injection of AAV9.ChR2-mCherry or unilateral ventricular injection of AAV9. eGFP-Cre and contralateral cortical injection of AAV9.ChR2-mCherry were used for this experiment. Fluorescence of GFP-positive soma and mCherry-labeled axons were visualized using a fluorescent mercury lamp (Excelitas Technologies Corp.). Slices containing clearly labeled mCherry+ axons from contralateral barrel cortex were used for recording. Slices with somatic infection of ChR2-mCherry due to virus leaked from contralateral hemisphere were discarded to avoid contamination from local inputs. The infection rate of recorded Cre-GFP+ neurons in barrel cortex was 3–5% (*Figure 2—figure supplement 1*). Only neurons residing in an area with densely labeled callosal axons were subject to recording to achieve a reliable magnitude and reduced variability of light-induced responses. For all LED experiments, responses were evoked by a 2 ms flash from a digitally controlled blue LED (final beam diameter: 350 µm; power: 0.1–4.6 mW; wavelength: 470 nm; M470L4-C1, Thorlabs Inc) through a 40 X water-immersed objective. The LED flash was centered on soma and proximal apical dendrites of recorded neurons.

To measure LED-evoked EPSCs, each cell (or cell pair) was stimulated 3–10 times with 20–30 s intervals. LED power was adjusted to obtain an EPSC amplitude of 100–1000 pA in WT neurons (for

cell pairs). The external solution for both LED and sCRACM experiments contained ACSF with 1 μM TTX, 100 μM 4-aminopyridine (4-AP), 10 μM (±)–3-(2-carboxypiperazin-4-yl)propyl-1-phosphonic acid (CPP) and 100 μM picrotoxin to isolate monosynaptic AMPAR-mediated excitatory inputs. Callosal input strength was measured as the peak amplitude of an average EPSC of 3–10 evoked EPSCs for each neuron.

To measure $Sr^{2+}$ evoked quantal events, each cell pair was stimulated with an LED flash every 30 s, 12–30 times in ACSF containing 4 mM $MgCl_2$, 4 mM $SrCl_2$, 10 μM CPP and 100 μM picrotoxin. Events occurring within a 1 s window prior to the LED were defined as spontaneous events and those occurring 50–350 ms post-LED were defined LED-evoked events. The frequency and amplitude of events were analyzed using MiniAnalysis (Synaptosoft).

NMDAR-mediated EPSCs from callosal inputs were pharmacologically isolated in ACSF containing: 3 mM $CaCl_2$, 0.1 mM $MgCl_2$, 20 μM DNQX, 20 μM glycine, 100 μM picrotoxin, 1 μM TTX, 100 μM 4-AP. An LED flash was delivered every 30 s, 9 times. Evoked EPSCs were averaged and peak amplitude was measured.

## Subcellular channelrhodopsin-assisted circuit mapping (sCRACM) and action potential dependent activation of ChR2

After collection of LED-evoked responses, the objective was switched to 4 X to visualize a broader area. ChR2 expressing axons were then stimulated by a blue laser (1 ms; wavelength: 473 nm; power range: 0.7–16 mW; final beam diameter: 25 μm; CrystaLaser) scanning through a 12 × 12 grid (50 μm spacing) in a pseudorandom order to avoid repeated activation of neighboring locations. The grid was aligned along the pia and centered in the medial-lateral position on the recorded somas. The grid was repeatedly scanned 2–4 times at 40 s intervals. Laser-evoked EPSCs were collected from each grid spot. Most neuron pairs ( > 90%) were homogeneously distributed between 150 and 300 μm from pia surface.

For action potential-dependent callosal synaptic strength measurements, an independent cohort of mice was used. The recording ACSF was similar to sCRACM experiments, except TTX and 4-AP were omitted, and divalent cations were increased (4 mM $MgCl_2$; 3 mM $CaCl_2$) and CPP was added to reduce polysynaptic activation of local circuits. For L2/3 neurons, an 8 × 8 grid with 75 × 100 μm x-y spacing was used. For L5 neurons, an 8 × 8 grid with 75 × 125 μm x-y spacing was used.

## Analysis of blue laser evoked EPSCs

For sCRACM, EPSCs evoked from two to four laser-stimulation at each grid spot from an individual neuron were averaged. The peak amplitude of the average EPSC (between 5–80 ms after laser onset) was determined as the input strength for that spot. A spatial map of input strengths was then generated for each individual neuron. The spatial maps for all neurons of each genotype were then aligned to soma location, oriented with respect to the pial surface, and averaged to generate an average spatial map of input strengths for each genotype aligned to the soma. A color representation of the average spatial map for each genotype were generated for the figures. Pixel size for the genotype-averaged color maps was halved through pixel interpolation (25 × 25 μm) to provide better spatial resolution for soma alignment. Vertical profiling of input strength was achieved by averaging inputs from each horizontal row and plotting against the vertical distance from soma. Input strength from a specific area was calculated by averaging input strengths within an area for each neuron and averaging according to genotype.

For action-potential-dependent ChR2 activation and circuit mapping, analysis was performed as for sCRACM, except EPSC amplitudes were averaged during 2–30 ms after laser onset to exclude contamination from polysynaptic responses.

## Laser scanning photostimulation (LSPS) with glutamate uncaging

LSPS experiments were performed similar to that described previously (*Rajkovich et al., 2017*; *Shepherd et al., 2003*). For all experiments, only brain slices with L2/3 apical dendrites parallel to the slice surface were used to ensure preservation of the planar barrel cortical geometry of cross-layer synaptic pathways spanning at least 3 barrel columns. Usually 2–4 brain slices per animal met such criteria. ACSF included 4 mM $MgCl_2$, 4 mM $CaCl_2$, and CPP (10 μM) to reduced polysynaptic local circuit activity (*Rajkovich et al., 2017*; *Shepherd et al., 2003*), and 4-Methoxy-7-nitroindolinyl-caged

-L-glutamate, MNI glutamate (MNI, 0.3 mM, either Tocris-1490 or HelloBio-HB0423). A 1 ms UV laser flash (wavelength: 355 nm; power range: 30–40 mW; final beam diameter: 20 μm; DPSS Lasers Inc) was delivered at individual points within a 16 × 16 grid (50 × 60 μm x-y spacing) in a pseudorandom order. The grid was aligned along the pia surface and centered medial-laterally on the soma location. The entire grid was repeatedly scanned 2–4 times at 40 s intervals. Two to four maps were acquired for each neuron included in all datasets.

For each neuron, a single average map was calculated from acquired LSPS maps, where at each stimulation point the averaged light evoked EPSC area was calculated within a time window of 5–80 ms following the laser pulse. If a response was observed within 5 ms of LSPS and displayed kinetics visibly faster than the longer-latency EPSC then it was considered to be a non-synaptic, 'direct' response to uncaging of glutamate on the recorded neuron. Direct responses were removed from the map and not included in analysis. For responses with a major component of monosynaptic transmission and minor contamination from 'direct' activation, 'direct' response component was subtracted from the whole response by fitting a double-exponential decay equation. An IR-DIC image of the slice with patch pipettes in place and stimulation grid was acquired prior to LSPS for marking soma location and anatomical features of the slice (i.e. barrels, etc.). Finally, a color map for each neuron was created. All individual color maps within genotype were then overlaid upon spatial alignment with respect to the center of the 'home' barrel directly beneath recorded L2/3 neurons. Superimposition of the average maps was achieved by (1) transposing each map such that the home barrel center was located at the origin of alignment grid, (2) preserving the medial-lateral orientation of the brain slice, and (3) stretching the home barrel in x and y dimensions to normalize barrel size. Pixel size for the genotype-averaged color maps was halved through pixel interpolation (25 × 30 μm) to provide better spatial resolution for soma alignment. Black pixels within an averaged color map in the figures indicate deleted direct responses or pixels that did not meet the minimum sampling threshold (minimum of n = 8 neurons per stimulation point).

## Whisker trimming

Unilateral whisker trimming was performed daily on mice starting at P15 until the day before experiment (P23-30) using a miniature electric shaver. All whiskers from the trimmed pad were maintained at a length <1 mm. For a litter of mice, half of the mice would be trimmed on the right whisker pad, ipsilateral to the AAV9.ChR2-mCherry injection (postsynaptic deprivation), while the other half would be trimmed on the left whisker pad, contralateral to the AAV9.ChR2-mCherry injection (presynaptic deprivation).

## Statistics

All the statistical tests and graphs were performed using GraphPad Prism 8 (GraphPad Software Inc). Prior to statistical tests of significance, each dataset underwent normality tests (D'Agostino-Pearson and Anderson-Darling) to determine if parametric or non-parametric tests should be used. All statistical tests are two-sided. Only datasets tested to be normally distributed by both tests were subject to parametric statistics – unpaired or paired t-test. Otherwise, Mann-Whitney test or Wilcoxon matched pairs test were used as indicated in the figure legends. For data collected from double-patch experiments, paired statistics were used as each pair of neurons would be considered correlated. For comparing more than two groups of data, two-way ANOVA or mixed effect analysis with multiple comparisons was used. Repeated measures ANOVA was used for vertical profile maps in *Figures 1 and 2*.

## Acknowledgements

We thank Patricia Hahn, Jacob Eli Bowles, and Christopher Williams for technical assistance with the mice. JRG, KMH, ZZ, conceived various aspects of the project. ZZ and JRG performed experiments and analyzed data. ZZ and KMH wrote the manuscript with input from all authors. This work was supported by grants from the NIH grants to KMH and JRG (R01 HD052731, U54HD082008, U54HD104461) and JRG (R03MH104366).

## Additional information

### Funding

| Funder | Grant reference number | Author |
|---|---|---|
| National Institutes of Health | R01 HD052731 | Jay R Gibson<br>Kimberly M Huber |
| National Institutes of Health | U54HD082008 | Jay R Gibson<br>Kimberly M Huber |
| National Institutes of Health | U54HD104461 | Jay R Gibson<br>Kimberly M Huber |
| National Institutes of Health | R03MH104366 | Jay R Gibson |

The funders had no role in study design, data collection and interpretation, or the decision to submit the work for publication.

### Author contributions

Zhe Zhang, Conceptualization, Data curation, Formal analysis, Visualization, Writing – original draft, Writing – review and editing; Jay R Gibson, Conceptualization, Data curation, Funding acquisition, Project administration, Software, Supervision, Writing – review and editing; Kimberly M Huber, Conceptualization, Funding acquisition, Methodology, Project administration, Supervision, Visualization, Writing – original draft, Writing – review and editing

### Author ORCIDs

Zhe Zhang http://orcid.org/0000-0003-2859-3859
Jay R Gibson http://orcid.org/0000-0002-6279-0736
Kimberly M Huber http://orcid.org/0000-0002-7479-0661

### Ethics

This study was performed in strict accordance with the recommendations in the Guide for the Care and Use of Laboratory Animals of the National Institutes of Health. All of the animals were handled according to approved institutional animal care and use committee (IACUC) protocols (2017-101986) of the University of Texas Southwestern Medical Center.

### Decision letter and Author response

Decision letter https://doi.org/10.7554/eLife.71555.sa1
Author response https://doi.org/10.7554/eLife.71555.sa2

## Additional files

### Supplementary files
• Transparent reporting form

### Data availability

All data generated or analyzed during this study are included in the manuscript and supporting files. Source data files have been provided for all figures and figure supplements.

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
