## [Decision Letter]

**Acceptance summary:**

The authors find that long-range interhemispheric synapses are selectively weakened following loss of function of the gene mediating fragile X syndrome, the most common inherited form of intellectual disability. Using clever genetic and physiological approaches in mice, the authors show that the effect is cell autonomous and occurs postnatally by impeding the normal developmental strengthening of these synapses. The results convincingly enhance our understanding of the complex pathophysiology of neurological dysfunction in this developmental disorder.

**Decision letter after peer review:**

Thank you for submitting your article "FMRP regulates experience-dependent maturation of callosal synaptic connections and bilateral cortical synchrony" for consideration by *eLife*. Your article has been reviewed by 3 peer reviewers, including Sacha B Nelson as the Reviewing Editor and Reviewer #1, and the evaluation has been overseen by Lu Chen as the Senior Editor.

Essential revisions:

1) Reviewers were concerned about the interpretation of the sparse conditional knockout experiments because of ambiguities in which cells and synapses were affected. Specifically, reviewer 2 felt, "the approach they used induces Fmr1 KO in the population of neurons in one hemisphere without control over the brain areas/layers/cell types within it. Thus, interpretation of data from experiments that involves the local circuit is difficult (i.e., local circuit connectivity/sensory deprivation), because Fmr1 KO is not restricted to post (L2/3) but also pre (e.g., L4) and potentially beyond (e.g. ipsilateral thalamus)" and reviewer 3 agreed with this concern. Both reviewers felt the concern could be addressed by demonstrating the sparseness of the labeling in a supplementary figure or additional panel.

2) All three reviewers felt that the EEG data were not compelling. This could be addressed either by removing these data or through additional experiments using the conditional KO and using the same cortical region studied in the rest of the manuscript.

3) The reviewers also felt that additional experiments were needed to address the question of whether the callosal deficits are lasting or represent a transient developmental defect that normalizes by adulthood, as was observed earlier in development with other synaptic deficits.

*Reviewer #1 (Recommendations for the authors):*

I have several suggestions for improvement that could be addressed either through additional experiments or through textual changes to acknowledge alternative interpretations.

1. The strontium and NMDA only experiments in figure 5 are suggestive but fall short of demonstrating a retention of silent synapses. The most direct way to demonstrate this would be to use minimal electrical or optogenetic stimulation and directly demonstrate the existence of silent synapses. However, there are aspects of this interpretation that do not make sense. With sparse deletion, callosal synaptic input was normal at P18-20 and synaptic loss occurred later (P23-30) but the silent synapse interpretation would require that a large number of callosal synapses remain silent until 3 weeks of age and then are strengthened by "AMPAfication." this is much later than silent synapses are thought to be lost in the neocortex, so this would be quite surprising. The experiments of figure 5 do not seem conclusive first because the effect on frequency (C) are quite modest relative to the large change in overall synaptic drive seen in prior experiments. A potential limitation of the NMDA only experiments is that these receptors may saturate masking presynaptic changes. Also, one would want to fully map the spatial profile of the response as with the AMPA responses. I think it would be fine to simply acknowledge the limitations of the evidence for this model in the discussion and admit that other factors might also contribute.

2. I was surprised that the discussion does not use the term "competition." The results seem most consistent with the idea that local and long range synapses normally compete and that biasing this competition with deprivation can normalize the bias introduced by the knockout if applied to the correct set of synapses. This is of course only a suggestion that the authors are free to use or not as they see fit. Another, more minor suggestion along these lines is to refer to old ultrastructural and anatomical studies (e.g. Czeiger and White 1993 and references therein) that callosal synapses are quite similar to other long-range intracortical synapses. This would help make the argument that although the callosal connection is useful to study physiologically or in vivo, the pathology likely extends to many other long range connections.

Discussion between the reviewers revealed that the other reviewers did not find this explanation likely because of the lack of increase in 4->2/3 synapses following manipulations that reduce callosal input. I think there may be a ceiling effect because the 2/3->4 projection is one of the strongest in the cortex, but wanted you to be aware that this view was not shared by the 3 reviewers.

3. Given the supplementary results confirming the effects seen by Bureau et al., but only in the germ-line knockout, it would seem important to acknowledge in the discussion that the in vivo synchrony results could reflect changes in local circuits and not only altered callosal function.

4. Finally, one small clarification would be helpful. The age ranges in Figures1 and 2 overlap. If the data in Figure 1 are analyzed by age, do they show the same effect as Figure 2? This needn't be the case, but it would be helpful to note the results and your thinking about this.

*Reviewer #2 (Recommendations for the authors):*

Use of AAV9:

AAV9 has a retrograde property (e.g. Haery et al., 2019), which could reduce the specificity of input sources. Some estimates of the retrograde transfection rate will clarify the impact of non-callosal inputs on the data.

LED stimulation method:

It is unclear photostimulation via 40x objective covers all the dendritic arbor of recorded neurons. Based on the information from Figure 1G scale bar, 200 μm LED spot size will only cover proximal dendrites, especially for deep L2/3 neurons.

Subdividing the age range for callosal input analysis in Figure 2:

This seems arbitrarily chosen. An explanation of why this is done is helpful.

The extent of neuronal labeling achieved by ventricular injection of AAV9.GFP-Cre at P1:

This needs to be clarified as Kim et al. (2013) cited used different AAV serotype/promoters (and age). The low mag image(s) showing the labeling pattern in the affected hemisphere (at least one covering all layers in S1) is helpful. Please clarify how 'infection rate… <10%' (line 495) is quantified.

Reporting distribution of recorded L2/3 neurons:

This information helps interpret the averaged sCRACM data. The length of apical dendrites varies with depth, with some L2 neurons having no clear/oblique apical dendrite. Including those into the average map (said to be aligned to the soma) will generate strong inputs at basal than distal. Please also clarify how dendritic morphology was verified to be intact (no cut) after the recording. How does the data look when maps are aligned at pia?

On sCRACM data:

It seems the sCRACM data do not significantly add much to the LED data or the conclusion. The authors could consider moving those to supplemental figures to make the presentation simple.

Reporting the distance between the pair of recorded neurons:

This clarifies their proximity and no edge effect affecting the data (which is a potential concern as the callosal axon occupies a narrow homotopic column (~200 μm based on Figure 1B)).

Inconsistency in the information:

The LED spot size is said to be 200 μm in the text but it is said to be 350 μm in the method.

Use of different dissection methods for a different age:

This raises the concern that the age-dependent effect could be due to slice quality.

*Reviewer #3 (Recommendations for the authors):*

1. It is indeed surprising that different excitatory synapses can show different phenotypes in the FMR1 KO resulting in both local hyperconnectivity (in L5) or delayed maturation but then ultimately normal function (at L4 to L2/3) – but then reduced function at callosal inputs. This recovery of function suggests that the maturation of callosal inputs might be delayed but ultimately normal in this model, and experiments to examine these inputs in older animals are required to interpret this result.

2. The deprivation studies are difficult to interpret. What is the mechanism behind this effect?

3. Interneuron function is altered in the FMR1 KO (Goel et al., 2018; Gibson et al., 2014). Unless the only (or primary) thing that is impaired in the KO is the strength of the callosal input, it will be hard to attribute the lack of coherence in gamma to this connection at a single timepoint. What if local inhibitory circuits are altered in the KO? Or, callosal inputs to interneurons?

4. Why look at callosal effects on synchronization in auditory function, when all measurements are done in S1/whisker deprivation?

5. Gamma coherence was tested at a different age. It is reasonable that the synaptic deficits observed at the single timepoint examined might be rectified at later ages, or they could possibly be exacerbated. These synaptic deficits might be some primary phenotype of FMR1 KO, but the correlated reduction in gamma coherence may be secondary to this; for example, reduced synaptic strength could drives elimination of callosal axons over time (especially since there are white matter deficits in autism patients), resulting in reduced coherence. I actually think it might be better to suggest that the reduction in callosal afferent strength could influence interhemispheric communication without presenting the last figure, that is not well-controlled and is less integrated with the data in the rest of the manuscript.

6. Example EEG traces that show this effect would greatly add to the interpretability of this analysis. Is overall EEG power different within a hemisphere? Is it altered equally in both sensory and motor areas?

7. There is no discussion of the molecular mechanisms of FMR1, or how it might differentially influence one type of synapse but not another. In addition, the overall hypothesis that they are testing appears to be that FMR1 suppresses long-range connections but enhances or does not influence local connections. To these ends, it would be useful to look at long-range connections that are not callosal – for example, from M1 or S2. Is there something special about callosal inputs or is it the timing of their maturation, for example?

---

## [Author Response]

Essential revisions:1) Reviewers were concerned about the interpretation of the sparse conditional knockout experiments because of ambiguities in which cells and synapses were affected. Specifically, reviewer 2 felt, "the approach they used induces Fmr1 KO in the population of neurons in one hemisphere without control over the brain areas/layers/cell types within it. Thus, interpretation of data from experiments that involves the local circuit is difficult (i.e., local circuit connectivity/sensory deprivation), because Fmr1 KO is not restricted to post (L2/3) but also pre (e.g., L4) and potentially beyond (e.g. ipsilateral thalamus)" and reviewer 3 agreed with this concern. Both reviewers felt the concern could be addressed by demonstrating the sparseness of the labeling in a supplementary figure or additional panel.

We understand the concerns on our sparse deletion paradigm and appreciate the suggestion from reviewers. We now include a new supplementary figure (Figure 2—figure supplement 1) showing representative low-and high magnification epifluorescence GFP images taken under experimental settings. The low-mag image shows the representative sparseness of Cre-GFP expression across all layers of barrel cortex and part of the striatum underneath cortex, which is consistent with the low amount of virus we applied (~5 x 10^8^ GC total per animal). We also include some representative high-mag images from L2/3, L4 and L5 in the same figure. Based on our experience, for a 0.05 mm^2^ field of view (FoV) at a specific focus under 40x objective, we find 0-3 transfected cells. As a simple estimation of the transfection rate, we counted the number of transfected cells and total number of neurons identified by IR-DIC for each FoV across 4 injected mice and calculated 3-5% of cells are virally transfected with Cre-GFP. Given the sparseness of Fmr1 deletion, deletion in presynaptic L4 neurons or other brain regions should be minimal and equally affect inputs to postsynaptic Fmr1 KO neurons and their WT neighbors. Therefore, with this sparse deletion paradigm our observations should reflect the function of postsynaptic Fmr1/FMRP.

2) All three reviewers felt that the EEG data were not compelling. This could be addressed either by removing these data or through additional experiments using the conditional KO and using the same cortical region studied in the rest of the manuscript.

We removed the EEG data from the manuscript and also revised the title to better reflect slice results. We have also removed authors that contributed the EEG data.

3) The reviewers also felt that additional experiments were needed to address the question of whether the callosal deficits are lasting or represent a transient developmental defect that normalizes by adulthood, as was observed earlier in development with other synaptic deficits.

To answer this question, we performed simultaneous recordings in 2-month old mice with neonatal sparse deletion of Fmr1 and contralateral callosal input labelling with ChR2. Similar to that observed in younger mice, we observe weak callosal input mediated EPSCs in postsynaptic Fmr1 KO L2/3 neurons as compared to WT neighbors. This result suggests that weak callosal synaptic inputs persist into young adulthood in the Fmr1 KO. These data are now included in the revised manuscript in Figure 2F-G.

Reviewer #1 (Recommendations for the authors):I have several suggestions for improvement that could be addressed either through additional experiments or through textual changes to acknowledge alternative interpretations.1. The strontium and NMDA only experiments in figure 5 are suggestive but fall short of demonstrating a retention of silent synapses. The most direct way to demonstrate this would be to use minimal electrical or optogenetic stimulation and directly demonstrate the existence of silent synapses. However, there are aspects of this interpretation that do not make sense. With sparse deletion, callosal synaptic input was normal at P18-20 and synaptic loss occurred later (P23-30) but the silent synapse interpretation would require that a large number of callosal synapses remain silent until 3 weeks of age and then are strengthened by "AMPAfication." this is much later than silent synapses are thought to be lost in the neocortex, so this would be quite surprising.

We agree with the reviewer that we did not demonstrate more silent synapses at Fmr1 KO callosal synaptic inputs and our results only suggest this possibility. We have revised our conclusions and discussion of these results to indicate that more experiments must be done to conclusively demonstrate silent synapses.

With regard to the time course of callosal synapse development, results in Figure 2E indicate that callosal synapses continue to strengthen between P18 and P30 in WT neurons, suggesting there may be ongoing “AMPAfication” at callosal synapses during the 3rd postnatal week. To our knowledge, there is not published work measuring developmental progress of silent synapses at callosal inputs. We also did not measure the developmental changes in NMDAR mediated callosal synaptic strength so we cannot conclusively determine if the strengthening is due to AMPAfication, additional synapses or other presynaptic mechanisms. We discuss this point briefly in the discussion.

The experiments of figure 5 do not seem conclusive first because the effect on frequency (C) are quite modest relative to the large change in overall synaptic drive seen in prior experiments. A potential limitation of the NMDA only experiments is that these receptors may saturate masking presynaptic changes. Also, one would want to fully map the spatial profile of the response as with the AMPA responses. I think it would be fine to simply acknowledge the limitations of the evidence for this model in the discussion and admit that other factors might also contribute.

We agree with the reviewer that the changes in event frequency in Sr^2+^ were modest (~20%) in comparison to the robust 40-50% decrease in evoked EPSC amplitudes (in Ca^2+^) in Fmr1 KO neurons. To further test the synaptic locus of change, we measured the coefficient of variance (C.V.) of LED-evoked callosal EPSCs onto WT and Fmr1 KO neurons (from experiments in Figs, 2D (P23-30) and Figure 5B_1_). C.V. is inversely proportional to release probability and synapse number (Manabe et al., 1993). We observed a significant (~20%) increase in CV in Fmr1 KO in comparison to neighboring WT neurons (p<0.05; Figure 3—figure supplement 1). This finding is consistent with the decrease in frequency of evoked events in Sr^2+^. While changes in CV and frequency are modest, together they indicate that the weakening of callosal mediated EPSCs in Fmr1 KO is in part a result of decreased synapse number and/or presynaptic release probability.

Thank you for pointing out the possibility of NMDAR saturation. We revised the discussion to acknowledge the limitations in interpretation of our results.

2. I was surprised that the discussion does not use the term "competition." The results seem most consistent with the idea that local and long range synapses normally compete and that biasing this competition with deprivation can normalize the bias introduced by the knockout if applied to the correct set of synapses. This is of course only a suggestion that the authors are free to use or not as they see fit. Another, more minor suggestion along these lines is to refer to old ultrastructural and anatomical studies (e.g. Czeiger and White 1993 and references therein) that callosal synapses are quite similar to other long-range intracortical synapses. This would help make the argument that although the callosal connection is useful to study physiologically or in vivo, the pathology likely extends to many other long range connections.Discussion between the reviewers revealed that the other reviewers did not find this explanation likely because of the lack of increase in 4->2/3 synapses following manipulations that reduce callosal input. I think there may be a ceiling effect because the 2/3->4 projection is one of the strongest in the cortex, but wanted you to be aware that this view was not shared by the 3 reviewers.

Thank you for this comment and we agree that local and long-range synapses may compete for connectivity. In the original manuscript, we discussed this in terms of a homeostasis between local and long-range inputs, but we have revised our discussion to include the possibility of a competition-based mechanism. Our results in L5 support this view. Postsynaptic deletion of Fmr1 KO in L5 neurons results in hyperconnectivity between local L5 neurons (Patel et al., 2014) but reduced callosal input strengths (Figure 4C,D). Less is known in L2/3. Postsynaptic deletion of Fmr1 KO in L2/3 neurons has no effect on L4 input strengths, but we don’t know if there is hyperconnectivity with L2/3 neighbors as there is in L5. Unfortunately, the very local, within L2/3 connectivity, is masked by direct responses in the LSPS maps and therefore labor-intensive quadruple simultaneous recordings of L2/3 pairs are required to get at this question as we performed previously in L5 and believe is beyond the scope of this current manuscript. It is possible that all “local” synapses may not be equal in their homeostasis or competition with long-range inputs. Interlayer local synapses (L4→L2/3) may be more resistant or stable, whereas within layer (L5→L5 or L2/3→L2/3) may be more plastic or susceptible to competition/homeostasis. These interesting possibilities are future directions of this work.

Thank you for informing us of the Czeiger and White work. We agree that Fmr1 may also promote other long-range cortico-cortical inputs and added this to the discussion.

3. Given the supplementary results confirming the effects seen by Bureau et al., but only in the germ-line knockout, it would seem important to acknowledge in the discussion that the in vivo synchrony results could reflect changes in local circuits and not only altered callosal function.

We have removed the EEG data from the manuscript, so we also removed the discussion text of the mechanisms related to the in vivo synchrony.

4. Finally, one small clarification would be helpful. The age ranges in Figures 1 and 2 overlap. If the data in Figure 1 are analyzed by age, do they show the same effect as Figure 2? This needn't be the case, but it would be helpful to note the results and your thinking about this.

We subdivided the data by age from global Fmr1 KO (Figure 1). At P18-20 in the global Fmr1 KO, we observe significant weakening of callosal synaptic inputs in the Fmr1 KO in comparison to WT (p< 0.05; WT; n=15; KO; n=7; Author response image 1). This contrasts with the postsynaptic cell autonomous deletion (Figure 2D, E), where we did not observe callosal synaptic weakening until after P23. In the discussion, we suggest that the discrepancy between global and postsynaptic Fmr1 KO may be due to the later, postnatal, AAV-Cre mediated, deletion of Fmr1 in the postsynaptic KO condition, likely about P7-P9 (Rajkovich et al., 2017). This may delay the onset of callosal synaptic weakening with postsynaptic Fmr1 KO, as compared to global germline deletion of Fmr1. Alternatively, there may be role for FMRP in presynaptic, callosal projecting neurons, or other cell types, such as oligodendrocytes within the first postnatal weeks to establish callosal synaptic connections.

**Author response image 1. sa2fig1:** Young (P18-20) global *Fmr1* KO mice have weak callosal synaptic connections. Left: Raw LED-induced EPSC amplitudes in WT and *Fmr1* KO animals (WT = 181.2 ± 22.18 pA, n = 15; KO = 106.1 ± 17.77 pA, n = 7, *p < 0.05, unpaired t-test); Right: LED-induced EPSC amplitude normalized to LED power and log transformed (WT = 2.216 ± 0.125, n = 15; KO = 1.718 ± 0.118, n = 7, *p < 0.05, unpaired t-test).

Reviewer #2 (Recommendations for the authors):Use of AAV9:AAV9 has a retrograde property (e.g. Haery et al., 2019), which could reduce the specificity of input sources. Some estimates of the retrograde transfection rate will clarify the impact of non-callosal inputs on the data.

For all recordings, we only performed experiments if we did not observe ChR2-mCherry expressing cell bodies in the recorded hemisphere. In a small minority of mice, we did observe many mCherry positive cells in the recorded hemisphere, which we ascribed to improper injection and virus leakage and did not record from these mice. Furthermore, for the >200 neuron recordings (majority in L2/3 and the rest in L5) we performed with blue light stimulation, we only observed a delayed synaptic current (latency ~2-5 msec) and never encountered a neuron with a fast onset (< 2 msec) blue-light induced current due to direct ChR2 activation. In addition, retrograde transport of AAV9 happens with high vector doses (~10^10^ GC per injection site), which is much higher than the amount of virus we applied (~10^8^-10^9^ GC total per animal). Also, based on literature, it mostly happens across bilateral hippocampi and among subcortical regions while is much less prominent across bilateral cortices (Cearley and Wolfe, 2006; Haery et al., 2019; Masamizu et al., 2011). This suggests with our virus injection conditions, we are getting expression in the injected hemisphere with little to no leak or retrograde transport of AAV9 ChR2-mCherry into the recorded hemisphere.

LED stimulation method:It is unclear photostimulation via 40x objective covers all the dendritic arbor of recorded neurons. Based on the information from Figure 1G scale bar, 200 μm LED spot size will only cover proximal dendrites, especially for deep L2/3 neurons.

We apologize for the mistake and confusion of the LED stimulation size. The correct size for LED under 40x objective should be 350 µm in diameter. We have changed this information in the main text.

Subdividing the age range for callosal input analysis in Figure 2:This seems arbitrarily chosen. An explanation of why this is done is helpful.

Subdivision of the age groups was based on the distribution of data (See Reviewer Figure 2). For the postsynaptic Fmr1 KO experiment in Figure 2, we initially decided to record between P18 and P30, because: (1) this matches the age range we tested in the constitutive germline Fmr1 KO (Figure 1 and 2) we wanted to match the age range of the age where we observed hyperconnectivity between L5 neurons (Patel et al., 2014). After collecting the data, we observed consistent weaker callosal inputs onto KO neurons at P23 and later (Author response image 2), whereas at earlier ages (P18-22), there was not a consistent effect. Based on these reasons, we reported the data in two groups (i.e. P18-20 and P23-30) and performed the follow-up experiments within the age range P23-30.

**Author response image 2. sa2fig2:** For each neuron pair, a ratio of *Fmr1* KO to WT responses was taken, log10 transformed and plotted by the postnatal day. A value less than 0 will indicate that *Fmr1* postsynaptic KO neuron has weaker callosal synaptic inputs than the neighboring WT neuron.

The extent of neuronal labeling achieved by ventricular injection of AAV9.GFP-Cre at P1:This needs to be clarified as Kim et al. (2013) cited used different AAV serotype/promoters (and age). The low mag image(s) showing the labeling pattern in the affected hemisphere (at least one covering all layers in S1) is helpful. Please clarify how 'infection rate… <10%' (line 495) is quantified.

We addressed this concern in “Essential Revisions”, point #1 (above) and in Figure 2—figure supplement 1. Both low magnification (all layers) and high magnification images are shown.

Reporting distribution of recorded L2/3 neurons:This information helps interpret the averaged sCRACM data. The length of apical dendrites varies with depth, with some L2 neurons having no clear/oblique apical dendrite. Including those into the average map (said to be aligned to the soma) will generate strong inputs at basal than distal. Please also clarify how dendritic morphology was verified to be intact (no cut) after the recording. How does the data look when maps are aligned at pia?

Recorded L2/3 neurons were homogeneously distributed within 150 – 300 μm from the pia surface, which should correspond to the range of L2 to mid-L3 (Hooks et al., 2011) (Author response image 3). This information has been updated in the Methods under sCRACM section. Neurons that reside closer to pia did lack the information for ‘distal’ responses. We agree that by averaging all maps including the ones in upper L2 may bias the averaged map to have more representation of peri-soma locations. However, even with the maps with ‘distal’ responses, the major difference was still seen in peri-soma/proximal dendritic locations because the ‘distal’ responses are usually small and thus harder to detect a difference. Aligning the sCRACM maps to pia instead of neuron soma, we still observe weak callosal input maps in Fmr1 KO neurons in general, and the averaged maps show decreasing gradient from the central ‘hotspot’ of response to the periphery.

**Author response image 3. sa2fig3:** Left: sCRACM maps from WT and *Fmr1* global KO (Figure 1) aligned to pia surface; Right: sCRACM maps from WT and postsynaptic *Fmr1* KO neuron pairs (Figure 2 – figure supplement 2) aligned to pia surface. Cyan dots represent the location of soma.

We didn’t perform neuron filling or reconstruction with our experiments for post-verification of dendritic morphology. Previously, by sparsely transfecting AAV-mCherry in cortex and following the primary dendrite of L5 and L2/3 pyramidal neurons in slice, we find that when the tissue block is perpendicular to the platform during cutting, the primary dendrites in barrel cortex will be largely parallel to the cutting surface of slice, i.e. there will be minimal cut-off for the primary dendrite. Also, for our experiment, we usually record at a depth around 50 μm below the cutting surface and visually follow and verify intact dendrites using IR-DIC of recorded neurons. With these precautions, we think the recorded L2/3 neurons should have a largely preserved dendritic tree.

On sCRACM data:It seems the sCRACM data do not significantly add much to the LED data or the conclusion. The authors could consider moving those to supplemental figures to make the presentation simple.

We have moved the sCRACM data from original Figures 2 and 5 to supplemental figures. We kept one example of sCRACM in Figure 1 to show a confirmation of the callosal synaptic weakening with a different method.

Reporting the distance between the pair of recorded neurons:This clarifies their proximity and no edge effect affecting the data (which is a potential concern as the callosal axon occupies a narrow homotopic column (~200 μm based on Figure 1B)).

During the experiment, we find the closest healthy WT pyramidal neuron next to the fluorescent (transfected) neuron at the same focal plane. Based on our experience and some high-mag pictures, the linear distance between the center points of the cell pair is usually 10-40 µm. This information has been updated in the Methods. As for the potential edge effect, since the relative positions of the KO and WT neurons to each other and the callosal column edge were random, this should automatically even out the possible bias. We also attempted to record neuron pairs, whenever possible, within the center of the labelled callosal axon column.

Inconsistency in the information:The LED spot size is said to be 200 μm in the text but it is said to be 350 μm in the method.

We apologize for the mistake and confusion of the LED stimulation size. The correct size for LED under 40x objective should be 350 µm in diameter. We have changed this information in the main text.

Use of different dissection methods for a different age:This raises the concern that the age-dependent effect could be due to slice quality.

We think slice quality/perfusion is unlikely to contribute to the age-dependent effect that we observed in Figure 2. First, we didn’t see any difference in terms of the resting membrane potential across age groups (see Author response image 4), suggesting that there are no depolarization/health issues with the neurons. Second, with the simultaneous recording paradigm we used, slice quality should have equal contribution to the WT and Fmr1 KO neurons. Then the only possibility would be that perfusion itself unmask the postsynaptic Fmr1 dependent effect. This is also unlikely as we see weak callosal inputs from global KO experiments (Figure 1) in which only a small percent (~15%) of mice were transcardially perfused.

**Author response image 4. sa2fig4:** 

Reviewer #3 (Recommendations for the authors):1. It is indeed surprising that different excitatory synapses can show different phenotypes in the FMR1 KO resulting in both local hyperconnectivity (in L5) or delayed maturation but then ultimately normal function (at L4 to L2/3) – but then reduced function at callosal inputs. This recovery of function suggests that the maturation of callosal inputs might be delayed but ultimately normal in this model, and experiments to examine these inputs in older animals are required to interpret this result.

In response to the reviewer, we performed additional experiments that are presented in Figure 2F and described in “Essential Revisions” point #3. Briefly, we observe that weak callosal synaptic inputs with neonatal, postsynaptic deletion of Fmr1 persists into young adulthood (P57-65; Figure 2F,G)

2. The deprivation studies are difficult to interpret. What is the mechanism behind this effect?

The reviewer is correct that sensory deprivation has multiple effects on brain development that make the specific mechanisms behind this effect difficult to interpret. FMRP function is regulated by neuronal activity and glutamate activation of synapses and we wanted to know if manipulating neuronal activity in the neocortex, using sensory deprivation, would affect or interact with Fmr1-dependent regulation of callosal synaptic inputs. Our sensory deprivation experiments suggest that the postsynaptic L2/3 Fmr1 KO neuron must be active for callosal synaptic weakening to occur. We do not know the mechanisms by which Fmr1 and sensory-driven activity interact but suggest that this could be due to enhanced activity-induced long-term synaptic depression (LTD) in Fmr1 KO neurons (in normal experience) or alternatively a homeostatic upscaling of callosal synaptic strengths in the sensory deprivation condition. Future experiments will investigate these possibilities.

3. Interneuron function is altered in the FMR1 KO (Goel et al., 2018; Gibson et al., 2014). Unless the only (or primary) thing that is impaired in the KO is the strength of the callosal input, it will be hard to attribute the lack of coherence in gamma to this connection at a single timepoint. What if local inhibitory circuits are altered in the KO? Or, callosal inputs to interneurons?

We agree with the reviewer that we cannot conclude that reduced interhemispheric gamma coherence in the Fmr1 KO is only due to reduced callosal synaptic strength, especially given the other known changes in Fmr1 KO circuitry. However, based on results from callosal cutting experiments, reduced coherence in gamma and beta would be a predicted if callosal synaptic connectivity was weak. Callosal inputs to inhibitory neurons may also be changed in the Fmr1 KO and this is something we are interested in exploring in the future.

4. Why look at callosal effects on synchronization in auditory function, when all measurements are done in S1/whisker deprivation?5. Gamma coherence was tested at a different age. It is reasonable that the synaptic deficits observed at the single timepoint examined might be rectified at later ages, or they could possibly be exacerbated. These synaptic deficits might be some primary phenotype of FMR1 KO, but the correlated reduction in gamma coherence may be secondary to this; for example, reduced synaptic strength.

The ISPC EEG data was in hand of our collaborators at UC Riverside and was consistent with our S1 results. However, we agree with the reviewer that this is not a logical comparison so we removed the EEG data from the manuscript.

Could drives elimination of callosal axons over time (especially since there are white matter deficits in autism patients), resulting in reduced coherence. I actually think it might be better to suggest that the reduction in callosal afferent strength could influence interhemispheric communication without presenting the last figure, that is not well-controlled and is less integrated with the data in the rest of the manuscript.

Again, we agree and have removed the EEG data from the manuscript.

6. Example EEG traces that show this effect would greatly add to the interpretability of this analysis. Is overall EEG power different within a hemisphere? Is it altered equally in both sensory and motor areas?

Because we have removed the EEG figure, these results are no longer relevant to the manuscript.

7. There is no discussion of the molecular mechanisms of FMR1, or how it might differentially influence one type of synapse but not another. In addition, the overall hypothesis that they are testing appears to be that FMR1 suppresses long-range connections but enhances or does not influence local connections. To these ends, it would be useful to look at long-range connections that are not callosal – for example, from M1 or S2. Is there something special about callosal inputs or is it the timing of their maturation, for example?

We have added a brief discussion of potential molecular mechanisms by which FMRP may differentially regulate synaptic inputs (lines 376-383). This could be due to localized expression of dendritic mRNAs within basal or apical dendritic compartments in that are translationally regulated by FMRP to affect either L4 or callosal inputs, respectively. Alternatively, FMRP is known to translationally control some transsynaptic cell adhesion or scaffolding molecules that can recognize presynaptic inputs from a specific origin. We have not examined other long-range input pathways but would predict that these would be similarly weak in the Fmr1 KO. This is based on the reduced fMRI coherence of other long-range ipsilateral pathways (such as). Future experiments will explore these possibilities.

References:

Cearley, C.N., and Wolfe, J.H. (2006). Transduction characteristics of adeno-associated virus vectors expressing cap serotypes 7, 8, 9, and Rh10 in the mouse brain. Mol Ther 13, 528-537. 10.1016/j.ymthe.2005.11.015.

Haery, L., Deverman, B.E., Matho, K.S., Cetin, A., Woodard, K., Cepko, C., Guerin, K.I., Rego, M.A., Ersing, I., Bachle, S.M., et al. (2019). Adeno-Associated Virus Technologies and Methods for Targeted Neuronal Manipulation. Front Neuroanat 13, 93. 10.3389/fnana.2019.00093.

Hooks, B.M., Hires, S.A., Zhang, Y.X., Huber, D., Petreanu, L., Svoboda, K., and Shepherd, G.M. (2011). Laminar analysis of excitatory local circuits in vibrissal motor and sensory cortical areas. PLoS Biol 9, e1000572. 10.1371/journal.pbio.1000572.

Manabe, T., Wyllie, D.J., Perkel, D.J., and Nicoll, R.A. (1993). Modulation of synaptic transmission and long-term potentiation: effects on paired pulse facilitation and EPSC variance in the CA1 region of the hippocampus. J Neurophysiol 70, 1451-1459. 10.1152/jn.1993.70.4.1451.

Masamizu, Y., Okada, T., Kawasaki, K., Ishibashi, H., Yuasa, S., Takeda, S., Hasegawa, I., and Nakahara, K. (2011). Local and retrograde gene transfer into primate neuronal pathways via adeno-associated virus serotype 8 and 9. Neuroscience 193, 249-258. 10.1016/j.neuroscience.2011.06.080.

Patel, A.B., Loerwald, K.W., Huber, K.M., and Gibson, J.R. (2014). Postsynaptic FMRP promotes the pruning of cell-to-cell connections among pyramidal neurons in the L5A neocortical network. J Neurosci 34, 3413-3418. 10.1523/JNEUROSCI.2921-13.2014.

Rajkovich, K.E., Loerwald, K.W., Hale, C.F., Hess, C.T., Gibson, J.R., and Huber, K.M. (2017). Experience-Dependent and Differential Regulation of Local and Long-Range Excitatory Neocortical Circuits by Postsynaptic Mef2c. Neuron 93, 48-56. 10.1016/j.neuron.2016.11.022.